# Factor Graph-based Interpretable Neural Networks

**Yicong Li[1], Kuanjiu Zhou[1], Shuo Yu[1]\*, Qiang Zhang[1]\*, Renqiang Luo[2], Xiaodong Li[3], Feng Xia[3]**
[1]Dalian University of Technology, [2]Jilin University, [3]RMIT University

## Abstract

Comprehensible neural network explanations are foundations for a better understanding of decisions, especially when the input data are infused with malicious perturbations. Existing solutions generally mitigate the impact of perturbations through adversarial training, yet they fail to generate comprehensible explanations under unknown perturbations. To address this challenge, we propose AGAIN, a factor graph-based interpretable neural network, which is capable of generating comprehensible explanations under unknown perturbations. Instead of retraining like previous solutions, the proposed AGAIN directly integrates logical rules by which logical errors in explanations are identified and rectified during inference. Specifically, we construct the factor graph to express logical rules between explanations and categories. By treating logical rules as exogenous knowledge, AGAIN can identify incomprehensible explanations that violate real-world logic. Furthermore, we propose an interactive intervention switch strategy rectifying explanations based on the logical guidance from the factor graph without learning perturbations, which overcomes the inherent limitation of adversarial training-based methods in defending only against known perturbations. Additionally, we theoretically demonstrate the effectiveness of employing factor graph by proving that the comprehensibility of explanations is strongly correlated with factor graph. Extensive experiments are conducted on three datasets and experimental results illustrate the superior performance of AGAIN compared to state-of-the-art baselines[1].

## 1 Introduction

Comprehensibility of neural network explanations depends on their consistency with human insights and real-world logic. Comprehensible explanations promote better understanding of decisions and establish trust in the deployment of neural networks in high-stake scenarios, such as healthcare and finance (Virgolin & Fracaros, 2023; Fokkema et al., 2023; Luo et al., 2024a;b). However, as shown in Figure 1, interpretable neural networks are vulnerable to malicious perturbations which are infused into inputs, misguiding the model to generate incomprehensible explanations (Tan & Tian, 2023; Baniecki & Biecek, 2024). Such explanations may cause users to make wrong judgments, resulting in security concerns in high-stake domains. For example, a doctor prescribing medication based on a medically illogical explanation (i.e., incomprehensible explanation) of the pathological prediction may lead to misdiagnosis. Therefore, it is crucial to ensure that interpretable neural networks generate comprehensible explanations under perturbations.

Several existing efforts have been devoted to investigating comprehensible explanations (Kamath et al., 2024; Sarkar et al., 2021; Chen et al., 2019). Many of them craft adversarial samples by adding perturbations to the dataset beforehand and retrain the model with extra regularization terms. Regularization terms constrain model to generate explanations that are similar to the explanation labels of the adversarial samples. Empirical results show that retrained models are able to learn the adversarial sample data distribution and reduce the probability of being misled by the predetermined perturbation.

However, the above solutions assume perturbations are known to the model, which leads to their failure to generate comprehensible explanations under unknown perturbations (Novakovsky et al.,

---

\*Corresponding authors: `shuo.yu@ieee.org`, `zhangq@dlut.edu.cn`

[1]Source codes are available at `https://github.com/yushuowiki/AGAIN`.

2023). The reasons are as follows: 1) it is impossible to craft adversarial samples for all unknown perturbation types (Gürel et al., 2021); 2) even if the adversarial samples are available, retraining is effective for only a limited number of perturbation types simultaneously (Tan & Tian, 2023). Thus, despite recent progress on comprehensible interpretability, it is still challenging to provide comprehensible explanations under unknown perturbations. Considering this, we seek to solve this problem from a different perspective - instead of optimizing the training strategy, we innovate the inference process. Our goal is to design an interpretable neural network capable of rectifying incomprehensible explanations under unknown perturbations during inference. We draw inspiration from knowledge integration with factor graphs. Unknown perturbations cause the explanatory semantics to violate the exogenous knowledge in the factor graph (Tu et al., 2023; Xia et al., 2021). Factor graph reasoning enables us to identify and rectify explanatory logical errors without learning perturbations.

We propose AGAIN (fActor GrAph-based Interpretable neural Network), which generates comprehensible concept-level explanations based on the factor graph under unknown perturbations (Tiddi & Schlobach, 2022). AGAIN consists of three modules, including factor graph construction, explanatory logic errors identification, and explanation rectification. In the first module, semantic concepts, label catergories,

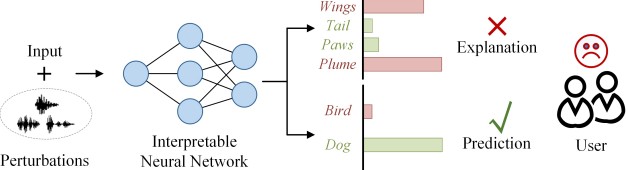

Figure 1: Interpretable neural networks suffer from perturbations that generate incomprehensible explanations. For instance, the model predicts the input as "Dog" but explains it with "Wings" and "Plume".

and logical rules between them are encoded as two kinds of nodes (i.e., variable and factor) in the factor graph, while their correlations are encoded as the edges. Based on the constructed factor graph, the logic relations among concepts and categories are explicitly represented. In the second module, AGAIN generates the concept-level explanations and predictive categories and then imports them into the factor graph to identify erroneous concept activations through logical reasoning. In the third module, we propose an interactive intervention switch strategy for concept activations to correct logical errors in explanations. The explanations that are further regenerated align with external knowledge. The regenerated explanations are used to predict categories. Extensive experiments are conducted on three datasets including CUB, MIMIC-III EWS, and Synthetic-MNIST. Experimental results demonstrate concept-level explanations generated by the proposed AGAIN under unknown perturbations have better comprehensibility compared to baselines such as ICBM, PCBM, free CBM, and ProbCBM.

Our contributions can be summarized as follows: 1) **against unknown perturbations:** we present an innovative interpretable neural network based on factor graph. It integrates real-world logical knowledge to generate comprehensible explanations under unknown perturbations; 2) **forward feedback:** we design logic error identification and rectification methods based on the factor graph. Our method is able to rectify logic violating explanations during inference without learning perturbations, unlike previous methods; 3) **theoretical foundation of factor graph:** we prove that the comprehensibility of explanations is positively correlated with the involvement of factor graph; 4) **superior performance:** we conduct extensive experiments on three datasets to demonstrate that AGAIN can generate more comprehensible explanations than existing methods under unknown perturbations.

## 2 RELATED WORK

**Comprehensible Explanation under Perturbation.** Studies of comprehensible explanations under perturbations can be divided into two categories: attacks on comprehensibility and defenses of comprehensibility. Studies of attacks on comprehensibility aim to design perturbations that misguide the model to generate incomprehensible explanations. Some methods modify salient mappings with perturbations that make the explanation incomprehensible to users (Ghorbani et al., 2019; Dombrowski et al., 2022). Furthermore, there are several efforts that propose additional types of perturbations (Rahmati et al., 2020; Carmichael & Scheirer, 2023; Huai et al., 2022). They demonstrate that many types of perturbations can undermine the comprehensibility of explanations. In contrast,

studies on defenses of comprehensibility aim to design defensive strategies to suppress the effects of perturbations on interpretations. These studies focus on adversarial training of interpretable neural networks so that the model generates comprehensible explanations despite perturbations. These studies are implemented in two ways. In the first approach, some methods annotate the adversarial samples with explanatory labels, and constrain the model to generate explanations similar to the labels (Boopathy et al., 2020; Lakkaraju et al., 2020; Chalasani et al., 2020). While promising, excessive adversarial training can easily lead to overfitting. In the second approach, some efforts further utilize different regularization terms based on adversarial training to mitigate overfitting, and allowing reasonable local shifts in explanations (Kamath et al., 2024; Sarkar et al., 2021; Chen et al., 2019). In addition, concept-based interpretable methods, which explain model decisions by generating a set of high-level semantic concepts, have gained great attention recently Koh et al. (2020b); Havasi et al. (2022); Wang et al. (2022). Moreover, it has been demonstrated that concept-based explanations can be erroneous and lose comprehensibility under perturbations Sinha et al. (2023). Meanwhile, they verify that retraining is effective in enhancing the comprehensibility of concept-based explanations. However, all the above methods assume that the perturbation is known to the model. Thus, how to improve the comprehensibility of the explanation under unknown perturbations remains open.

**Knowledge Integration with Factor Graph.** There have been extensive studies on knowledge integration with factor graphs (Tian et al., 2024; Gürel et al., 2021; Yang et al., 2022). These studies typically utilize factor graph reasoning to assemble predictions from multiple ML models. When one model predicts incorrectly, the factor graph can combine the exogenous knowledge to correct the error based on the predictions of other models. Empirical evidence suggests that integration of exogenous knowledge in factor graphs contributes to the predictive accuracy of ML models. In this paper, instead of improving predictive accuracy, we explore the possibility of using exogenous knowledge to guide interpretable neural networks for generating comprehensible explanations.

# 3 NOTATIONS AND PRELIMINARIES

**Interpretable Neural Network.** Interpretable neural networks are defined as neural networks that automatically generate explanations for decisions (Esterhuizen et al., 2022; Rieger et al., 2020; Peng et al., 2024). For more comprehensible explanations, we utilize a concept bottleneck model to generate concept-level explanations, which utilize various semantic concepts to explain the predictions Koh et al. (2020b); Huang et al. (2024). Specifically, let $x$ denote an input sample, the concept bottleneck model predicts the category $y$ and outputs a boolean vector $\mathbf{c} \in \{0,1\}^M$ of $M$ concepts. Let $c \in \mathbf{c}$ denote a concept. Let $c = 1$ indicate that concept $c$ is present in $x$ and influences the model decision. $\mathbf{c}$ is the concept-level explanation of the model prediction.

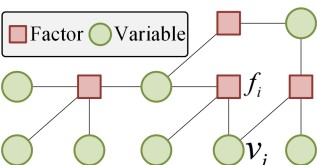

Figure 2: An example of the factor graph. It consists of 4 factors and 8 variables.

**Factor Graph.** Factor graph serves as a probabilistic graphical model to depict relationships among events (Yu et al., 2023; Bravo-Hermsdorff et al., 2023). As shown in Figure 2, within a factor graph, two node types exist: 1) variables, which delineate events; 2) factors, which articulate the relationships between events. Formally, a factor graph $\mathcal{G} = (\mathcal{V}, \mathcal{F})$ contains the set of variables $\mathcal{V}$ and the set of factors $\mathcal{F}$. We denote the set of edges as $\mathcal{E}$. For any $v_i \in \mathcal{V}$ and $f_i \in \mathcal{F}$, we let $(v_i, f_i) \in \mathcal{E}$ denote an edge of $\mathcal{G}$. Let $\mathcal{N}(f_i) = \{v_i \in \mathcal{V} | (v_i, f_i) \in \mathcal{E}\}$ denote the set of neighbors of factor $f_i$ in $\mathcal{G}$. We let variables correspond to concept and category labels. We let factors correspond to logical rules. This enables $\mathcal{G}$ to encode logical rules between concepts and categories.

**Known and Unknown Perturbation.** Formally, let $\delta$ denote perturbations uniformly. The designer of the model crafts adversarial samples against one perturbation $\delta_k$ to obtain a retrained model $h$ that minimizes $\|h(x; \theta) - h(x + \delta_k; \theta)\|$. $\theta$ is the model parameter. For model $h$, $\delta_k$ denotes one known perturbation, and any $\delta_u \in \{\delta | \delta \neq \delta_k\}$ denotes one unknown perturbation.

**Adversarial Attacks against Concept-level Explanations.** Unlike standard adversarial attacks, adversarial attacks against explanations do not compromise task predictions. For concept-level in-

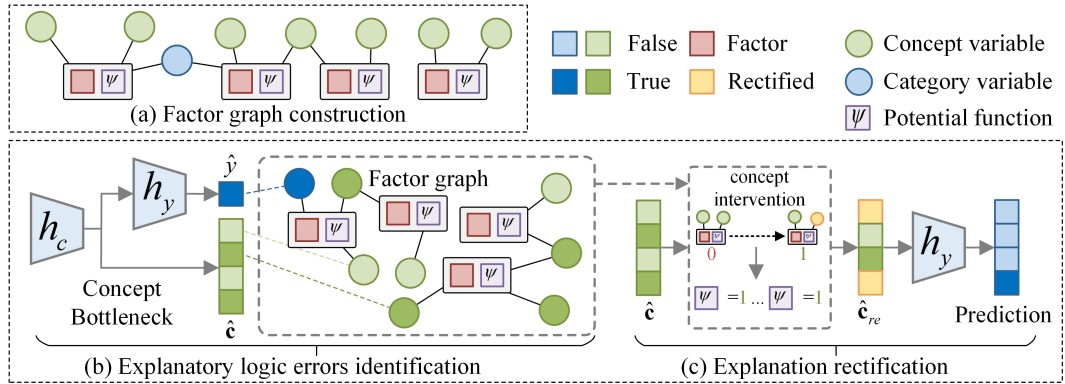

Figure 3: Overall structure of AGAIN.

terpretable models, such attacks can be categorized into three types: erasure attacks, introduction attacks, and confounding attacks (Sinha et al., 2023). 1) Erasure attacks: the goal of the erasure attacks is to subtly remove concepts from a concept-level explanation; 2) introduction attacks: the goal of introduction attacks is to allow the existence of irrelevant concepts; 3) confounding attacks: the goal of confounding attacks is to simultaneously remove relevant concepts and introduce irrelevant concepts. These attacks are technically simpler to implement (see the Appendix E for implementation details of these attacks).

## 4 THE DESIGN OF AGAIN

AGAIN consists of three modules: 1) first, we encode logic rules of the real-world as a factor graph (Section 4.1); 2) then, we generate the initial concept-level explanation through the concept bottleneck. The factor graph reasoning is utilized to identify whether the explanation of concept bottleneck violates the external logic, and thus to detect whether the perturbation exists (Section 4.2); 3) finally, an interactive intervention strategy is designed to rectify the explanation and input it to the category predictor (Section 4.3). The overall architecture of AGAIN is shown in Figure 3.

### 4.1 FACTOR GRAPH CONSTRUCTION

To construct a factor graph, we first define the logic rule set $\mathcal{R} = \{r_i\}_{i=1}^{N}$, which contains two types: 1) concept-concept rule: all predicates consist of concepts. Such rules are used to constrain potential relationships among various concepts. For instance, there is a rule of coexistence or exclusion between concepts $c_i$ and $c_j$, which can be formalized in logical notation as: $c_i \Leftrightarrow c_j$ or $c_i \oplus c_j$.

2) Category-concept rule: all predicates are defined by concepts and categories. Such rules are used to constrain potential correlations between concepts and categories. For instance, the coexistence or exclusion rule that exists between concept $c_i$ and category label $y_j$ can be formalized as: $c_i \Leftrightarrow y_j$ or $c_i \oplus y_j$.

Then, we encode the above logic rules into a factor graph $\mathcal{G}$. As shown in Figure 4, we illustrate the construction of the factor graph. Specifically, there are two types of variables $\mathcal{V} = \mathcal{V}^c \cup \mathcal{V}^y$, where $\mathcal{V}^c$ and $\mathcal{V}^y$ denote concept and category variable set, respectively, and are linked by $\mathcal{F}$. In this way, each factor $f_i \in \mathcal{F}$ corresponds to the $i$-th logic rule $r_i$. Each factor is defined as a potential function that performs logical operations based on different rules, which can be categorized into coexistence and exclusion operations.

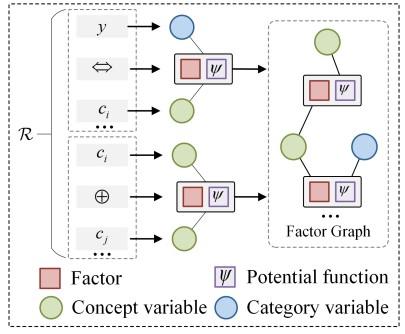

Figure 4: Factor graph construction.

Moreover, we define a potential function $\psi_i$ for each factor $f_i$, which outputs a boolean value for each $\mathcal{N}(f_i)$. If $\mathcal{N}(f_i)$ makes $r_i$ true, $\psi_i(\mathcal{N}(f_i)) = 1$, otherwise $\psi_i(\mathcal{N}(f_i)) = 0$.

For convenience, we denote $\psi_i(\mathcal{N}(f_i))$ as $\psi_i$. We define the weight $w_i \in [0,1]$ to represent the confidence level of $f_i$ in two methods, i.e., prior setting and likelihood estimation (The details can be referred to the Appendix C.5). Higher $w_i$ indicates that the logic rules of $f_i$ are more important for reasoning, and vice versa.

## 4.2 EXPLANATORY LOGIC ERRORS IDENTIFICATION

AGAIN generates an initial concept-level explanation and identifies logical errors in the initial explanation. Specifically, we employ a concept bottleneck structure, a popular concept-level interpretable module, to capture the semantic information from instances, which can learn the mapping between semantic information and concepts (Koh et al., 2020a). The concept bottleneck contains a concept predictor $h_c : \mathbb{R}^D \to \mathbb{R}^M$ and a category predictor $h_y : \mathbb{R}^M \to \mathbb{R}$. The instance $x$ is first mapped to the concept space by $h_c$ and obtains the corresponding concept activation vector, i.e., $\hat{\mathbf{c}} = h_c(x)$, where $\hat{\mathbf{c}} \in [0,1]^M$, and then the conceptual activation vector is fed into $h_y$ to yield the final predicted category, i.e., $\hat{y} = h_y(\hat{\mathbf{c}})$, where $\hat{y} \in \{0,1\}$. $\hat{\mathbf{c}}$ is defined as the initial explanation.

Next, $\mathcal{G}$ takes $\hat{\mathbf{c}}$ and $\hat{y}$ as inputs to assign $\mathcal{V}^c$ and $\mathcal{V}^y$, respectively. If concept $\hat{c} > 0.5, \hat{c} \in \hat{\mathbf{c}}$, we set variable $v_{\hat{c}} = 1, v_{\hat{c}} \in \mathcal{V}^c$, otherwise $v_{\hat{c}} = 0$. For the category variables, we set $v_{\hat{y}} = 1, v_{\hat{y}} \in \mathcal{V}^y$, and $\mathcal{V}^y \setminus v_{\hat{y}} = \{0\}^{K-1}$.

Subsequently, we evaluate the likelihood of the variable assignment under rule constraints through logical reasoning. Firstly, after each variable (concept and category) in $\mathcal{G}$ is assigned a value, boolean values are output from potential functions of all factors. These boolean values indicate whether the assignments of concepts and categories satisfy the logical rules represented by potential functions. Therefore, the weighted sum of all potential functions quantifies the extent to which concept assignments satisfy the logic rules in $\mathcal{G}$.

Then, we seek to obtain the likelihood of the current concept assignments occurring, conditional on the known categories and logic rules. We quantify this likelihood by computing a conditional probability using the weighted sum of potential functions. We consider all possible concept assignments and compute the expectation of current concept assignments. This expected value is considered as the conditional probability, which is then used to detect whether concept activations are perturbed. For illustrative purposes, we provide an example. Suppose there are concepts $A$ and $B$. The current concept assignment is $\{1,0\}$ denoting $A = 1$ (active) and $B = 0$ (inactive). We iterate through all four possible assignments: $\{1,0\}, \{0,1\}, \{1,1\}, \{0,0\}$. We compute the weighted sum of the potential functions for each of the four cases and compute the expectation of the potential function for $\{1,0\}$. This expectation is the conditional probability that concept assignment $\{1,0\}$ occurs conditionally on the known categories and logic rules. Formally, we denote this conditional probability as $\mathbb{P}(\mathcal{V}^c | \mathcal{V}^y)$:

$$\mathbb{P}(\mathcal{V}^c | \mathcal{V}^y) = \exp\left(\sum_{i \in N} w_i \psi_i\right) \bigg/ \sum_{\tilde{\mathcal{V}}^c \in \Phi} \left(\exp\left(\sum_{i \in N} w_i \psi_i\right)\right), \tag{1}$$

where $\Phi$ represents all cases of concept assignments, and $\tilde{\mathcal{V}}^c$ represents a case in $\Phi$. This implies that the denominator of Eq. (1) is the normalized constant term. We use $\mathbb{P}(\mathcal{V}^c | \mathcal{V}^y)$ to evaluate the comprehensibility of explanation $\hat{\mathbf{c}}$. Higher $\mathbb{P}(\mathcal{V}^c | \mathcal{V}^y)$ indicates that $\hat{\mathbf{c}}$ is more comprehensible, and vice versa. In theory, we consider that a comprehensible $\hat{\mathbf{c}}$ should satisfy each $r_i \in \mathcal{R}$, ensuring that $\mathbb{P}(\mathcal{V}^c | \mathcal{V}^y)$ attains an upper bound denoted as $_\vee\mathbb{P}(\mathcal{V}^c | \mathcal{V}^y)$:

$$_\vee\mathbb{P}(\mathcal{V}^c | \mathcal{V}^y) = \frac{1}{a} \max\left(\exp\left(\sum_{i \in N} w_i \psi_i\right)\right), \tag{2}$$

where $a$ denotes the denominator of Eq. (1). However, in practice, even concept explanations generated in a benign environment (without perturbations) rarely satisfy all the rules. Overly strict logical constraints may instead cause $\mathcal{G}$ to lose its ability to recognize perturbations. Therefore, we allow a comprehensible explanation to violate some low-weight rules. Naturally, we establish a relaxed identification condition for distinguishing explanations corrupted by perturbations from comprehensible explanations:

$$\mathbb{P}(\mathcal{V}^c | \mathcal{V}^y) > \partial \cdot {}_\vee\mathbb{P}(\mathcal{V}^c | \mathcal{V}^y), \tag{3}$$

where $\partial \in [0, 1]$ is a hyperparameter controlling the relaxation. $\partial$ approximate 1 implies a stricter constraint imposed by $\mathcal{G}$ on the explanation. If $\mathcal{V}^c$, $\mathcal{V}^y$ satisfies Eq. (3), then $\hat{\mathbf{c}}$ is comprehensible; otherwise, it is recognized as having logical error under perturbation. Further, we demonstrate theoretically that $\mathcal{G}$ contributes to comprehensible explanations. For a detailed theoretical analysis, please refer to Appendix B.

### 4.3 EXPLANATION RECTIFICATION

Once explanations with logical errors are identified, AGAIN rectifies the explanation and put it as an input to the category predictor for the final prediction. For this objective, we propose an interactive intervention switch strategy aimed at enhancing the conditional probability of the $\mathcal{G}$. The proposed strategy intervenes on the values of $\mathcal{V}^c$ and interactively observing the potential function difference. In this paper, we assume that $\hat{y}$ are unaffected under perturbations, thus we do not intervene in $\mathcal{V}^y$. The pseudocode of the interactive intervention switch is listed in Appendix A.

Our intervention strategy can be divided into three steps. First, we traverse all factors with $\psi_i = 1$. For factor $f_i \in \mathcal{F}$, we modify the boolean value of its concept variables, considering the modification as a single intervention operation. Given that $f_i$ may be connected to multiple concept variables, there exist numerous intervention cases. For instance, consider $f_i$ containing concept variables $v_i$ and $v_j$. There are three possible intervention cases: intervene only $v_i$, intervene only $v_j$, and intervene both $v_i$ and $v_j$. We define the full set of possible intervention cases for $f_i$ as $\mathcal{T}_i$. For each case $t_i \in \mathcal{T}_i$, we compute the potential function difference $s_i$, which represents the change in the potential function after executing $t_i$. Note that $t_i$ does not only change $f_i$, but also changes the 1-hop neighbor factors of $\mathcal{N}(f_i)$. Thus, we define $s_i$ as follows:

$$s_i = \sum_{j \in |\mathcal{F}_i|} w_j \left( \psi_j^{t_i} - \psi_j \right), \tag{4}$$

where $\mathcal{F}_i = \{ f_j \,|\, \mathcal{N}(f_i) \in \mathcal{N}(f_j) \} \cup \{ f_i \}$. $\psi_j^{t_i}$ denotes the $\psi_j$ value after $t_i$ intervention. Subsequently, after traversing through all possible interventions in $\mathcal{T}_i$, we identify the intervention with the highest $s_i$ as a candidate intervention. We generate the candidate intervention for each factor with $\psi_i = 1$. We aggregate all the candidate interventions into a final intervention, denoted as $t_*$. We execute $t_*$ on $\mathcal{V}^c$. From the set of intervened $\mathcal{V}^c$ and $t_*$, we generate a binary concept intervention vector $\mathbf{z} \in \{-1, 1\}^M$ and a binary mask vector $\mathbf{m}_{t_*} \in \{0, 1\}^M$. $\mathbf{z}$ denotes the concept activation status, where -1 indicates activated, and 1 indicates inactivated. $\mathbf{m}_{t_*}$ denotes whether the concept is intervened or not, where 1 indicates intervened, and 0 indicates not intervened.

Finally, we employ $\mathbf{z}$ and $\mathbf{m}_{t_*}$ to rectify the initial explanation $\hat{\mathbf{c}}$. We utilize $\mathbf{m}_{t_*}$ to aggregate $\hat{\mathbf{c}}$ and $\mathbf{z}$ for a rectified concept activation vector $\hat{\mathbf{c}}_{re}$:

$$\hat{\mathbf{c}}_{re} = \mathbf{z} \odot \mathbf{m}_{t_*} + \hat{\mathbf{c}} \odot \mathbf{m}'_{t_*}, \tag{5}$$

where $\mathbf{m}'_{t_*}$ is obtained by flipping the bits of $\mathbf{m}_{t_*}$. $\odot$ denotes dot product operation. The purpose of employing the intervention mask is to facilitate $\hat{\mathbf{c}}_{re}$ to retain activations in $\hat{\mathbf{c}}$ that are not intervened.

## 5 EXPERIMENTS

### 5.1 EXPERIMENTAL SETTINGS

**Datasets and Baselines.** We evaluated AGAIN on two real-world datasets, CUB, MIMIC-III EWS, and one synthetic dataset, Synthetic-MNIST. We choose two categories of methods. (1) Concept-level methods: CBM (Koh et al., 2020a), Hard AR (Havasi et al., 2022), ICBM (Chauhan et al., 2023), PCBM (Yuksekgonul et al., 2023), ProbCBM (Kim et al., 2023), Label-free CBM (Oikarinen et al., 2023), ProtoCBM (Huang et al., 2024), and ECBMs (Xu et al., 2024). (2) Knowledge integration methods: DeepProblog Manhaeve et al. (2018), MBM Patel et al. (2022), C-HMCNN Giunchiglia & Lukasiewicz (2020), LEN Ciravegna et al. (2023), DKAA Melacci et al. (2021), and MORDAA Yin et al. (2021). In addition, we compare AGAIN with the retrained versions of these baselines that employ state-of-the-art adversarial training strategy. More details on the datasets and baselines are provided in Appendix C.1 and C.2. The experimental results on the synthetic dataset are presented in Appendix D.4.

**Evaluation Metrics and Implementation Details.** To evaluate the performance of AGAIN, we use five metrics: predictive accuracy (P-ACC), explanatory accuracy (E-ACC), logical satisfaction metric (LSM), identification rate (IR), and success rate (SR). Higher scores indicate better performance for all metrics. Detailed descriptions of each metric are given in Appendix C.3. Additionally, the implementation details of AGAIN are provided in Appendix C.4.

## 5.2 EXPERIMENTAL RESULTS ON REAL-WORLD DATASETS

**Identifying Perturbations.** We apply adversarial perturbations acquired during black-box training to randomly perturb multiple instances in the test set. Known and unknown perturbations are denoted by $\delta_k$ and $\delta_u$, respectively, with $\epsilon$ representing the perturbation magnitude. We evaluate the ability of AGAIN to recognize logical errors of explanations by reporting SR and IR values under different perturbation magnitudes in Table 1. The results demonstrate that AGAIN achieves remarkable IR and SR values under both $\delta_k$ and $\delta_u$. Specifically, AGAIN attains nearly 100% IR across all perturbation magnitudes. With SR results averaging up to 98%, we also validate that factor graph $\mathcal{G}$ can effectively identify explanations from benign instances and permit them to directly predict categories without logical reasoning. Furthermore, it is also worth noting that as the perturbation magnitude increases, the IR value also gets larger. This observation is attributed to the larger perturbation magnitude causing a more pronounced logical violation in the generated explanations. The $\mathcal{G}$ more readily identifies these violations.

Table 1: IR and SR on two real-world datasets.

| Dataset | Metrics | Clear | $\delta_k$ | | | | $\delta_u$ | | | |
|---|---|---|---|---|---|---|---|---|---|---|
| | | | $\epsilon$=4 | $\epsilon$=8 | $\epsilon$=16 | $\epsilon$=32 | $\epsilon$=4 | $\epsilon$=8 | $\epsilon$=16 | $\epsilon$=32 |
| CUB | IR | - | 97.3(1.3) | 98.9(0.4) | 98.9(0.8) | 99.3(1.0) | 97.3(0.5) | 97.2(0.8) | 97.5(1.1) | 98.3(0.9) |
| | SR | 98.7(0.4) | 98.0(1.2) | 98.7(0.4) | 97.7(0.6) | 97.2(0.5) | 97.4(0.7) | 97.2(1.1) | 96.8(0.9) | 97.7(1.1) |
| MIMIC-III EWS | IR | - | 97.4(0.2) | 98.3(0.3) | 99.7(0.2) | 99.8(0.1) | 98.3(0.4) | 99.5(0.1) | 100.0(0.0) | 100.0(0.0) |
| | SR | 100.0(0.0) | 98.91(0.4) | 99.3(0.1) | 98.7(0.4) | 98.2(1.1) | 99.3(0.3) | 97.1(0.3) | 98.6(0.4) | 99.4(0.1) |

**Comprehensibility of Explanations.** To investigate the comprehensibility of the explanations generated by AGAIN, we perform extensive experiments on both datasets for evaluating the LSM of the explanations, and the comparison are reported in Table 2. The baselines subjected to the retraining are identified by the "-AT" suffix. The results reveal that the comprehensibility of the explanations generated by AGAIN outperforms all concept-level methods, including the "-AT" versions of these baselines, under different perturbation magnitudes. Particularly, previous interpretable models fail to generate logically complete explanations with LSMs lower than 48 under unknown perturbations of magnitude 32, but explanations from AGAIN can reach as high as 92.30. Moreover, we demonstrate that AGAIN is hardly affected by the perturbation magnitude compared to the baseline methods. This effect is attributed to the corrective capability provided by $\mathcal{G}$ for any level of logic violation. For kowledge integration methods, since DeepProblog, MBM, and C-HMCNN are unable to generate concepts, we splice their knowledge integration modules onto the CBM. The results show that the LSM of AGAIN is optimal. In contrast, deepProblog can only constrain category predictions, not concept predictions, which results in low LSM under perturbation. The knowledge introduced by methods MBM and C-HMCNN can constrain concepts, but they only use logical rules between concepts and concepts, making their performance inferior to AGAIN. Meanwhile, since DKAA and MORDAA have Multi-label predictors, we directly use Multi-label predictors to predict concepts. LEN can only constrain category predictions. DKAA and MORDAA detect adversarial perturbations in the samples using external knowledge, but they cannot correct the wrong concepts triggered by these perturbations.

**Validity of the Factor Graph.** As the theoretical analysis in Appendix B demonstrates, $\mathcal{G}$ improves the comprehensibility of explanations. We experimentally validate this claim and further demonstrate that increasing the number of factors in $\mathcal{G}$ enhances the predictive accuracy of concepts. Specifically, we employ subgraph $\mathcal{G}'$ extracted from the original $\mathcal{G}$ for reasoning and analyze the impact on prediction accuracy by increasing the ratio of $\mathcal{G}'$ to $\mathcal{G}$. In Figure 5, we depict the changes in P-ACC and E-ACC across four perturbation magnitudes on both datasets. It is evident that both P-ACC and E-ACC exhibit substantial improvement as the number of factors in $\mathcal{G}'$ increases. This observation indicates that $\mathcal{G}$ contributes in generating explanations with similarity to

Table 2: Comparisons of LSM for AGAIN with other concept-level interpretable baselines.

| Dataset | Method | clear | $\delta_k$ | | | | $\delta_u$ | | | |
|---|---|---|---|---|---|---|---|---|---|---|
| | | | $\epsilon=4$ | $\epsilon=8$ | $\epsilon=16$ | $\epsilon=32$ | $\epsilon=4$ | $\epsilon=8$ | $\epsilon=16$ | $\epsilon=32$ |
| CUB | CBM | 96.3(2.2) | 89.2(3.5) | 77.4(2.8) | 53.7(1.3) | 39.4(5.4) | 89.3(3.1) | 77.4(5.5) | 53.1(3.8) | 39.4(5.3) |
| | Hard AR | 85.6(0.9) | 77.3(1.2) | 66.4(3.2) | 50.8(1.8) | 47.6(0.8) | 77.4(0.5) | 66.2(2.6) | 50.4(1.4) | 47.2(1.8) |
| | ICBM | 95.4(1.3) | 86.4(0.8) | 77.3(0.4) | 56.4(1.6) | 39.5(1.5) | 86.7(0.7) | 77.6(2.4) | 56.9(1.9) | 39.8(3.2) |
| | PCBM | 95.7(1.6) | 85.6(1.0) | 76.3(1.5) | 58.7(2.1) | 40.3(1.2) | 87.5(1.5) | 78.1(1.4) | 57.6(1.8) | 41.6(2.6) |
| | ProbCBM | **96.8(0.4)** | 85.5(0.2) | 77.6(1.1) | 59.7(1.5) | 39.7(1.3) | 87.7(1.2) | 77.4(1.0) | 57.4(1.4) | 40.3(1.2) |
| | Label-free CBM | 96.4(1.3) | 84.3(1.7) | 76.9(1.4) | 58.5(2.1) | 40.8(0.3) | 87.3(1.6) | 77.8(1.2) | 57.3(1.4) | 40.0(1.8) |
| | ProtoCBM | 97.3(0.2) | 92.3(1.3) | 84.5(1.6) | 64.5(3.2) | 54.3(1.3) | 92.4(1.1) | 84.4(1.9) | 64.5(2.4) | 54.1(3.6) |
| | ECBMs | 96.4(1.3) | 92.1(1.5) | 87.5(3.7) | 70.4(2.8) | 64.7(2.1) | 92.2(0.9) | 86.6(3.7) | 70.4(2.9) | 64.4(6.1) |
| | CBM-AT | 93.4(1.4) | **92.9(1.3)** | 85.6(0.7) | 75.6(1.7) | 59.6(1.6) | 87.6(0.6) | 77.5(1.0) | 53.3(1.9) | 39.7(1.5) |
| | Hard AR-AT | 82.5(0.3) | 78.6(0.5) | 76.6(1.6) | 69.9(1.3) | 60.5(1.7) | 77.6(0.6) | 65.2(1.4) | 51.9(1.1) | 47.5(2.1) |
| | ICBM-AT | 91.5(1.3) | 91.6(2.4) | 86.3(1.9) | 79.3(1.6) | 70.6(1.5) | 80.4(1.2) | 77.6(2.1) | 56.4(1.8) | 39.4(1.6) |
| | PCBM-AT | 93.6(0.6) | 92.5(1.6) | 84.4(0.9) | 76.5(1.3) | 70.9(1.9) | 80.4(1.5) | 75.3(1.2) | 55.6(1.6) | 41.6(2.1) |
| | ProbCBM-AT | 93.5(0.9) | 90.3(1.4) | 83.3(1.3) | 78.2(1.6) | 70.3(1.8) | 87.4(0.9) | 77.6(1.2) | 57.5(1.6) | 40.6(1.4) |
| | Label-free CBM-AT | 93.4(1.3) | 91.4(0.8) | 86.2(1.4) | 80.9(1.6) | 78.5(1.5) | 87.8(1.4) | 77.4(1.8) | 57.7(1.7) | 41.8(2.9) |
| | ProtoCBM-AT | 94.4(0.7) | 92.7(1.1) | 87.2(1.9) | 70.3(4.2) | 68.7(2.7) | 91.7(1.2) | 82.5(1.5) | 60.2(2.4) | 52.1(6.1) |
| | ECBMs-AT | 93.6(1.0) | 91.9(2.5) | 88.1(2.1) | 83.1(3.8) | 78.4(3.4) | 90.7(2.4) | 83.7(2.5) | 68.7(2.7) | 66.7(3.7) |
| | LEN | 96.4(0.8) | 89.1(3.4) | 77.8(1.6) | 56.7(1.2) | 40.4(1.3) | 89.1(3.4) | 77.8(1.6) | 56.7(1.2) | 40.4(1.3) |
| | DKAA | 96.2(1.1) | 91.2(1.5) | 85.6(1.6) | 76.9(1.3) | 73.7(5.3) | 91.2(1.5) | 85.6(1.6) | 76.9(1.3) | 73.7(5.3) |
| | MORDAA | 96.5(0.1) | 91.7(1.5) | 86.1(1.8) | 80.6(2.1) | 76.8(3.1) | 91.7(1.5) | 86.1(1.8) | 80.6(2.1) | 76.8(3.1) |
| | DeepProblog | 96.4(0.2) | 89.2(3.5) | 77.4(2.8) | 53.7(1.3) | 39.4(5.4) | 89.2(3.5) | 77.4(5.5) | 53.1(3.8) | 39.4(5.4) |
| | MBM | 96.2(0.2) | 93.5(3.1) | 90.3(2.4) | 88.7(3.2) | 85.7(6.7) | 93.5(3.2) | 90.3(2.7) | 88.7(3.2) | 85.7(6.7) |
| | C-HMCNN | 96.5(0.4) | 93.6(7.2) | 89.7(1.2) | 87.6(2.5) | 85.0(3.2) | 93.6(7.2) | 89.7(1.2) | 87.6(2.5) | 85.0(3.2) |
| | AGAIN | 96.3(0.5) | 92.4(1.2) | **93.1(2.3)** | **93.8(1.9)** | **91.5(1.7)** | **94.5(1.6)** | **93.3(1.7)** | **93.8(1.4)** | **92.1(2.1)** |
| MIMIC-III EWS | CBM | 95.7(0.2) | 90.4(1.7) | 75.7(1.3) | 50.4(1.5) | 39.8(1.4) | 90.7(0.9) | 75.7(1.3) | 50.9(1.4) | 30.7(1.5) |
| | Hard AR | **96.7(0.3)** | 78.8(1.5) | 69.6(1.3) | 53.8(1.7) | 45.3(1.6) | 77.4(1.8) | 65.3(1.3) | 53.8(3.2) | 47.9(2.8) |
| | ICBM | 95.6(0.4) | 86.5(1.3) | 75.0(1.6) | 56.8(2.1) | 39.3(3.2) | 86.5(1.7) | 77.6(1.4) | 56.7(2.1) | 30.7(1.9) |
| | PCBM | 96.1(0.2) | 86.5(1.4) | 73.0(1.2) | 53.8(2.5) | 44.2(2.6) | 88.4(1.2) | 78.7(1.4) | 57.8(2.2) | 32.6(2.5) |
| | ProbCBM | 96.1(0.1) | 84.6(1.4) | 76.6(1.6) | 56.9(1.3) | 39.7(3.1) | 86.8(1.4) | 76.9(3.1) | 57.4(3.5) | 40.3(4.0) |
| | Label-free CBM | 96.1(0.1) | 86.5(1.2) | 76.5(1.6) | 65.5(2.1) | 40.3(2.3) | 86.9(0.9) | 77.6(4.1) | 56.8(6.3) | 42.3(7.4) |
| | ProtoCBM | 96.7(0.6) | 87.6(1.4) | 81.4(1.0) | 76.4(1.4) | 70.8(2.4) | 87.4(1.2) | 81.7(1.1) | 76.3(2.1) | 69.4(2.4) |
| | ECBMs | 97.9(0.2) | 88.4(1.2) | 79.4(1.3) | 70.8(2.6) | 65.6(3.2) | 88.6(2.7) | 79.4(2.1) | 70.4(5.4) | 66.1(4.8) |
| | CBM-AT | 94.2(0.4) | 90.3(1.2) | 85.4(1.3) | 78.8(1.9) | 60.9(1.9) | 85.7(2.5) | 77.5(2.3) | 50.8(3.1) | 40.8(3.7) |
| | Hard AR-AT | 94.2(0.7) | 88.4(1.4) | 76.9(1.6) | 70.2(2.6) | 65.3(3.1) | 77.5(1.1) | 62.1(1.1) | 50.7(1.1) | 44.2(1.1) |
| | ICBM-AT | 92.3(0.4) | 90.3(2.1) | 86.3(1.9) | 86.5(2.7) | 71.1(3.5) | 86.5(2.6) | 77.6(4.7) | 58.7(6.8) | 39.4(9.3) |
| | PCBM-AT | 94.2(0.6) | 90.3(1.7) | 84.4(3.2) | 78.9(3.4) | 69.2(4.7) | 81.7(3.7) | 75.3(3.3) | 54.6(4.1) | 42.3(4.9) |
| | ProbCBM-AT | 93.0(0.4) | 91.8(1.4) | 88.4(1.5) | 78.8(3.6) | 73.0(3.8) | 86.5(1.4) | 76.9(4.8) | 56.3(7.4) | 40.6(12.8) |
| | Label-free CBM-AT | 94.2(0.6) | 89.5(1.7) | 86.7(1.8) | 84.3(3.7) | 77.3(3.8) | 84.2(1.4) | 78.9(2.5) | 56.7(4.6) | 44.2(7.9) |
| | ProtoCBM-AT | 94.5(1.2) | 89.7(1.1) | 81.4(1.0) | 76.4(1.4) | 70.8(2.4) | 80.4(1.2) | 76.7(4.1) | 66.3(2.1) | 61.3(5.4) |
| | ECBMs-AT | 93.9(0.6) | 89.8(4.2) | 82.3(2.1) | 75.4(2.4) | 70.6(2.8) | 79.6(2.7) | 74.2(6.2) | 69.2(6.7) | 65.9(6.5) |
| | LEN | 96.4(0.6) | 90.3(2.6) | 75.7(2.8) | 50.3(3.8) | 40.2(7.1) | 90.3(2.6) | 75.7(2.8) | 50.3(3.8) | 40.2(7.1) |
| | DKAA | 96.5(1.5) | 96.1(1.6) | 87.3(2.7) | 79.8(2.1) | 75.8(4.7) | 96.1(1.6) | 87.3(2.7) | 79.8(2.1) | 75.8(4.7) |
| | MORDAA | 95.2(1.4) | 95.9(2.4) | 93.7(2.3) | 86.3(1.8) | 79.4(4.6) | 95.9(2.4) | 93.7(2.3) | 86.3(1.8) | 79.4(4.6) |
| | DeepProblog | 95.5(1.3) | 90.4(1.7) | 75.7(1.3) | 50.4(1.5) | 39.8(1.4) | 90.4(1.7) | 75.7(1.3) | 50.4(1.5) | 39.8(1.4) |
| | MBM | 95.9(0.5) | 92.7(4.2) | 88.7(2.4) | 86.3(1.3) | 84.4(5.7) | 92.7(4.2) | 88.7(2.4) | 86.3(1.3) | 84.4(5.7) |
| | C-HMCNN | 96.6(1.2) | 94.0(2.6) | 92.5(1.6) | 85.5(5.7) | 83.5(5.2) | 94.0(2.6) | 92.5(1.6) | 85.5(5.7) | 83.5(5.2) |
| | AGAIN | 96.1(0.3) | **96.1(0.7)** | **94.2(1.4)** | **96.1(1.2)** | **94.2(1.2)** | **94.0(2.7)** | **94.1(6.3)** | **94.2(2.4)** | **92.3(4.7)** |

Figure 5: The impact of the factor graph size on P-ACC and E-ACC across 4 perturbation magnitudes on two real-world datasets.

the ground truth explanations for improving the predictive accuracy. Moreover, as the number of factors in $\mathcal{G}'$ exceeds that of $\mathcal{G}$ (ratio $> 1.0$), E-ACC begin to converge. This also validates the setting for the number of factors in the original $\mathcal{G}$ is reasonable. In addition, we report the P-ACC and E-ACC comparison results of AGAIN on the CUB dataset (see Table 3 and Table 4). The comparison results of P-ACC and E-ACC on other datasets are provided in Appendix D.5. The results indicate that AGAIN is optimal for E-ACC on all three datasets. Furthermore, since perturbations do not impact the final predictions, the P-ACC remains consistent across different levels of perturbation. The P-ACC of AGAIN are comparable to the other baselines because the factor graph does not improve the task predictive accuracy. We present a comparison of E-ACC and P-ACC between the CBM and AGAIN on the two real-world datasets in Figure 6. The results show that AGAIN achieves higher

E-ACC values and comparable P-ACC compared to CBM under $\epsilon = 32$ perturbation. In the benign environment, E-ACC and P-ACC of AGAIN also remain comparable. The results suggest that factor graph logical reasoning does not affect prediction accuracy in the absence of perturbation.

Table 3: Comparisons of E-ACC between AGAIN and baselines on CUB.

| Method | clear | $\delta_k$ | | | | $\delta_u$ | | | |
|---|---|---|---|---|---|---|---|---|---|
| | | $\epsilon$=4 | $\epsilon$=8 | $\epsilon$=16 | $\epsilon$=32 | $\epsilon$=4 | $\epsilon$=8 | $\epsilon$=16 | $\epsilon$=32 |
| CBM-AT | 97.6(1.2) | 91.4(1.3) | 90.1(1.2) | 86.9(1.4) | 85.4(2.1) | 90.6(0.6) | 89.5(1.2) | 87.7(2.1) | 80.4(1.4) |
| Hard AR-AT | 97.5(1.4) | 93.1(0.4) | 90.4(1.6) | 86.3(1.7) | 84.7(2.5) | 91.2(0.2) | 86.7(1.5) | 84.1(2.3) | 79.6(2.1) |
| ICBM-AT | 96.8(1.1) | 94.0(2.1) | 89.7(2.4) | 85.8(2.3) | 84.8(3.2) | 89.3(2.7) | 85.1(2.1) | 80.5(3.1) | 78.9(1.2) |
| PCBM-AT | 97.8(1.1) | 92.4(2.7) | 89.4(1.2) | 86.4(1.9) | 83.5(2.4) | 92.4(1.6) | 89.8(3.1) | 85.3(2.6) | 80.4(1.5) |
| ProbCBM-AT | 96.9(1.0) | 92.5(1.0) | 90.2(1.4) | 84.6(2.3) | 83.9(2.1) | 90.6(3.2) | 88.6(1.6) | 82.5(1.7) | 79.4(1.3) |
| Label-free CBM-AT | 97.9(0.9) | 93.1(1.1) | 92.1(2.3) | 86.7(1.6) | 84.7(3.2) | 90.4(2.1) | 86.7(2.1) | 85.6(1.2) | 80.5(2.8) |
| ProtoCBM-AT | 98.1(0.7) | 93.1(1.3) | 90.4(1.5) | 85.4(3.1) | 83.9(2.4) | 91.6(1.8) | 86.8(3.2) | 83.1(1.3) | 81.5(3.1) |
| ECBMs-AT | **98.2(0.5)** | 92.8(2.4) | 89.3(3.4) | 86.6(2.2) | 83.7(2.5) | 91.3(1.3) | 89.4(1.3) | 84.8(2.5) | 80.7(1.9) |
| LEN | 97.7(0.2) | 93.1(1.5) | 89.4(1.4) | 86.1(1.4) | 79.9(3.7) | 93.1(1.5) | 89.4(1.4) | 86.1(1.4) | 79.9(3.7) |
| DKAA | 98.2(1.1) | 92.0(1.4) | 90.6(1.3) | 86.4(3.2) | 80.3(2.6) | 92.0(1.4) | 90.6(1.3) | 86.4(3.2) | 80.3(2.6) |
| MORDAA | 98.1(1.4) | 93.1(0.7) | 91.2(1.4) | 86.3(1.8) | 79.4(4.6) | 93.1(0.7) | 91.2(1.4) | 86.3(1.8) | 79.4(4.6) |
| DeepProblog | 96.9(0.8) | 91.0(2.3) | 89.8(1.1) | 85.2(1.3) | 80.2(1.4) | 91.0(2.3) | 89.8(1.1) | 85.2(1.3) | 80.2(1.4) |
| MBM | 97.9(1.0) | 91.6(1.2) | 90.3(1.4) | 86.3(1.3) | 82.7(3.2) | 91.6(1.2) | 90.3(1.4) | 86.3(1.3) | 82.7(3.2) |
| C-HMCNN | 98.1(0.2) | 91.3(1.4) | 91.6(2.1) | 85.5(5.7) | 84.5(2.3) | 94.0(2.6) | 92.5(1.6) | 85.5(5.7) | 83.5(5.2) |
| AGAIN | 97.5(0.1) | **94.1(0.2)** | **94.1(0.2)** | **93.8(0.4)** | **93.6(0.7)** | **94.9(0.3)** | **94.9(0.7)** | **93.5(1.1)** | **93.9(0.7)** |

Table 4: Comparisons of P-ACC between AGAIN and baselines on CUB.

| Method | clear | $\delta_k$ | | | | $\delta_u$ | | | |
|---|---|---|---|---|---|---|---|---|---|
| | | $\epsilon$=4 | $\epsilon$=8 | $\epsilon$=16 | $\epsilon$=32 | $\epsilon$=4 | $\epsilon$=8 | $\epsilon$=16 | $\epsilon$=32 |
| CBM-AT | 62.5(1.2) | 62.4(1.2) | 62.4(1.1) | 62.2(1.4) | 60.5(1.1) | 62.4(1.0) | 62.6(1.2) | 59.4(0.8) | 59.6(1.2) |
| Hard AR-AT | 62.4(1.3) | 62.2(0.6) | 61.2(0.6) | 61.6(0.4) | 59.6(0.8) | 61.6(1.2) | 60.3(0.3) | 59.4(0.6) | 59.7(1.2) |
| ICBM-AT | 62.4(1.7) | 62.2(0.6) | 61.3(0.6) | 61.8(1.0) | 59.3(1.1) | 62.4(1.1) | 60.2(0.2) | 59.5(0.4) | 59.8(1.2) |
| PCBM-AT | 62.5(0.4) | 62.1(0.6) | 61.0(0.6) | 61.6(0.3) | 59.6(0.8) | 62.6(1.2) | 61.0(0.1) | 59.4(0.7) | 59.5(1.1) |
| ProbCBM-AT | 62.4(0.5) | 62.2(0.4) | 61.4(1.1) | 61.6(0.4) | 59.5(0.8) | 61.6(1.2) | 60.6(0.3) | 59.3(1.1) | 59.6(0.8) |
| Label-free CBM-AT | 62.7(1.3) | 62.7(0.6) | 61.1(0.4) | 61.3(0.2) | 60.7(1.1) | 61.3(1.2) | 60.6(0.3) | 59.3(0.6) | 59.5(0.8) |
| ProtoCBM-AT | 62.4(1.3) | 62.1(0.7) | 61.3(0.8) | 61.6(0.3) | 59.2(0.7) | 61.4(1.0) | 60.1(1.2) | 59.4(0.9) | 59.5(1.2) |
| ECBMs-AT | 62.6(1.2) | 62.4(1.6) | 62.4(1.6) | 61.7(0.4) | 59.6(0.8) | 61.5(1.1) | 60.3(1.2) | 59.3(1.1) | 59.6(1.2) |
| LEN | 62.4(0.4) | 62.2(1.3) | 62.3(0.5) | 59.4(1.4) | 59.4(1.4) | 62.2(1.3) | 62.3(0.5) | 59.4(1.6) | 59.4(1.4) |
| DKAA | 61.4(1.5) | 62.6(1.2) | 62.4(0.5) | 59.4(1.2) | 59.2(1.7) | 62.6(1.2) | 62.4(0.5) | 59.4(1.2) | 59.2(1.7) |
| MORDAA | 62.2(0.1) | 62.5(0.8) | 61.8(1.3) | 59.5(1.1) | 59.3(0.9) | 62.5(0.8) | 61.8(1.3) | 59.5(1.1) | 59.3(0.9) |
| DeepProblog | 59.2(0.3) | 59.3(1.4) | 59.7(0.2) | 59.3(1.4) | 59.5(1.7) | 59.3(1.4) | 59.7(0.2) | 59.3(1.4) | 59.5(1.7) |
| MBM | 62.5(1.2) | 62.5(1.2) | 61.6(1.1) | 59.7(1.4) | 59.4(1.1) | 62.5(1.2) | 61.6(1.1) | 59.7(1.4) | 59.4(1.1) |
| C-HMCNN | 62.5(1.2) | 62.6(1.2) | 62.4(1.1) | 59.6(1.4) | 59.4(1.1) | 62.6(1.2) | 62.4(1.1) | 59.6(1.4) | 59.4(1.1) |
| AGAIN | 62.5(0.4) | 62.2(0.6) | 61.2(0.6) | 61.6(0.4) | 59.6(0.8) | 61.6(1.2) | 60.3(0.3) | 59.4(0.6) | 59.6(0.8) |

**Rectification of Interactive Intervention Switch.** In Figure 7(a) and 7(b), we illustrate several instances of two real-world datasets along with rectified explanation segments with a dimension of 5 and show the utilized rules. The interactive intervention switch effectively rectifies the logical error of the explanation based on the predefined rules, thereby enhancing the overall logical coherence of the explanation.

**Ablation Study.** We conducted ablation studies to examine the effectiveness for each rule type in $\mathcal{G}$. The larger number of rules in CUB compared to MIMIC-III EWS contributes to a more significant ablation effect, so we executed ablation studies on the CUB data. We investigated the performance of all factor graph variants by reporting the LSM results in Table 5. $\mathcal{F}^y$ and $\mathcal{F}^c$ denote the set

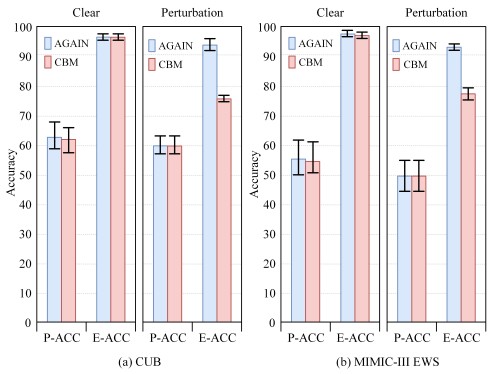

Figure 6: The comparison of P-ACC and E-ACC on the two real-world datasets.

of factors encoding category-concept rules and the set of factors encoding concept-concept rules, respectively.

According to Table 5, we can draw the following conclusions. First, the variant without $\mathcal{G}$ (i.e., $\mathcal{F} = \emptyset$) yielded the lowest LSM values, affirming the essential role of $\mathcal{G}$ in the model. Second, $\mathcal{G}$ encoding both concept-concept and category-concept rules (i.e., $\mathcal{F} = \mathcal{F}^y \cup \mathcal{F}^c$) achieved the best performance. This factor graph (constructed in this paper) demonstrates an average improvement of 6.86 over other variants of $\mathcal{G}$ that encode only the concept-concept or category-concept rules.

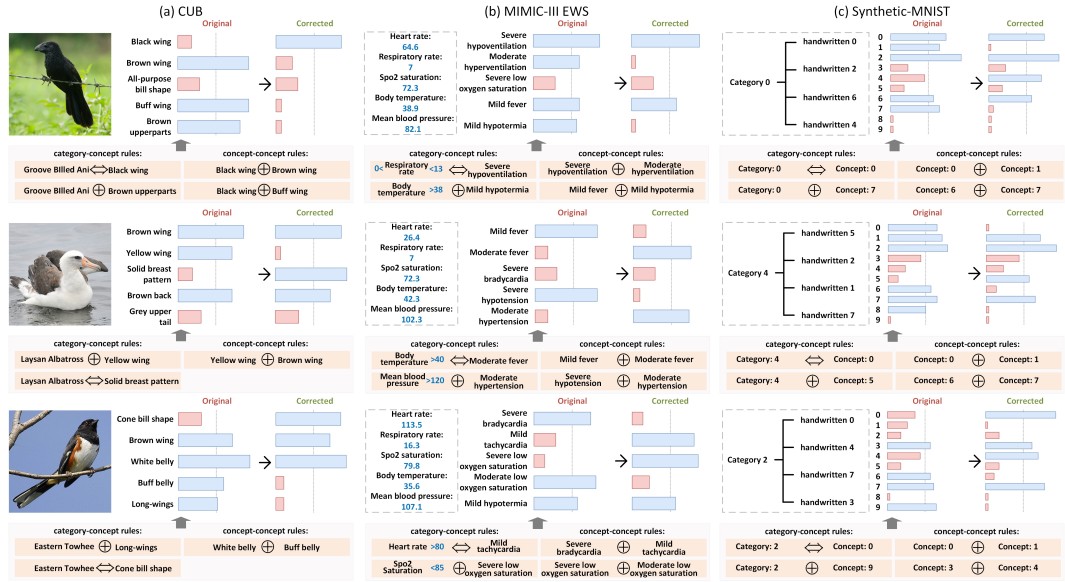

Figure 7: Rectified explanation results on three datasets. The bar represents the normalized activation values of the concepts. Blue bars indicate activated concepts and red bars indicate inactivated concepts. The orange area shows the logical rules followed.

This indicates that both rules are essential for comprehensibility of explanations. Finally, the variant containing only $\mathcal{F}^c$ performs lower than the variant containing only $\mathcal{F}^y$. This suggests that the category-concept rules contain more direct logical knowledge about category prediction.

Table 5: Ablation study on the CUB dataset: impact of rule types.

| Factor Set | $\epsilon = 4$ | $\epsilon = 8$ | $\epsilon = 16$ | $\epsilon = 32$ |
|---|---|---|---|---|
| $\mathcal{F} = \emptyset$ | 89.2(1.1) | 77.4(2.1) | 53.7(2.4) | 39.4(3.1) |
| $\mathcal{F} = \mathcal{F}^y$ | 92.5(0.3) | 89.5(2.1) | 85.5(2.2) | 84.0(2.5) |
| $\mathcal{F} = \mathcal{F}^c$ | 91.4 (0.4) | 86.8(2.4) | 82.6(2.3) | 80.1(2.3) |
| $\mathcal{F} = \mathcal{F}^y \cup \mathcal{F}^c$ | **94.5(0.3)** | **93.3(2.9)** | **93.8(2.2)** | **92.1(6.1)** |

## 6 CONCLUSION AND DISCUSSION

In this paper, we explore the comprehensibility of explanations under unknown perturbations and propose AGAIN, an factor graph-based interpretable neural network. Inspired by the knowledge integration of factor graphs, AGAIN obtains comprehensible explanations by encoding prior logical rules as the factor graph and utilizing factor graph reasoning to identify and rectify logical error in explanations. It addresses the inherent limitations of current adversarial training-based interpretable models by guiding explanation generation during inference. Furthermore, we provide a theoretical analysis to demonstrate that factor graphs significantly contribute to obtaining comprehensible explanations. We present an initial attempt to generate comprehensible explanations under unknown perturbations from the inference perspective. AGAIN provides an effective solution for the defense of interpretable neural networks against various perturbations and meanwhile saves the high cost of retraining. It takes a significant step towards resolving the crisis of trust between humans and interpretable models. In addition, we note two limitations of AGAIN: 1) the validity of AGAIN relies on the correct prediction categories. Wrong categories imported into the factor graph cause explanations to be wrongly rectified; 2) when domain knowledge changes, the factor graph needs to be reconstructed, which implies AGAIN lacks generalizability. We leave these for future work.

### ACKNOWLEDGMENTS

This paper is supported by the National Key R&D Program of China (Grant No. 2024YFA1012700). Thanks to Shan Jin and Ciyuan Peng for their help with data preprocessing.

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

## A ALGORITHMS

The overall algorithm for the interactive intervention switch is summarized in Algorithm 1.

---

**Algorithm 1:** Interactive intervention switch

---

**Input** : The factor graph $\mathcal{G}$, original concept activation vector $\mathbf{a}$
**Output:** Rectified concept activation vector $\mathbf{a}_{re}$

1 **for** $f_i$ **in** $\mathcal{F}$ **do**
2    **if** $\psi_i = 0$ **then**
3      Obtain the set $\mathcal{T}_i$ of all possible intervention cases for $f_i$;
4      **for** $t_i$ **in** $\mathcal{T}_i$ **do**
5        Compute $s_i$ through a single intervention $t_i$ according to Eq. (4);
6      Select $t_i$ with the highest $s_i$ as a candidate intervention;

7 Aggregate all candidate interventions into final intervention $t_*$;
8 Obtain intervention vector $\mathbf{z}$ and mask vector $\mathbf{m}_{t_*}$;
9 Aggregate $\mathbf{z}$ and $\mathbf{a}$ based on Eq. (5) yields the resulting $\hat{\mathbf{c}}_{re}$;
10 **return** $\hat{\mathbf{c}}_{re}$;

---

## B THEORETICAL ANALYSIS OF AGAIN

In this section, we present theoretical proofs to ensure that the $\mathcal{G}$ in AGAIN contributes to generating comprehensible explanations under unknown perturbations. Previous theoretical analyses on the validity of interpretable models have emphasized that comprehensible explanations tend to have a high similarity to explanatory labels (Li et al., 2020; Karpatne et al., 2017; Von Rueden et al., 2021). Therefore, we hope to guarantee that AGAIN can generate comprehensible explanations by proving that $\mathcal{G}$ contributes to explanations under unknown perturbations in approximation to the explanatory labels. Specifically, our theoretical proof is divided into two parts. In the first part, we establish correlations between the conditional probabilities of the factors and the lower bounds on the predictive accuracy of concepts. In the second part, we show that this lower bound increases according to the larger size of $\mathcal{G}$.

### B.1 THE CONCEPT PREDICTIVE ACCURACY LOWER BOUND FOR AGAIN

Under the adversarial distribution, given an perturbation $\delta$ and the concept explanation label $\mathbf{c}$, the accuracy of the explanation is denoted as $\mathcal{A}^h$:

$$\mathcal{A}^h := \prod_{c \in \mathbf{c}} \mathbb{P}_\delta \left( \hat{c} = c \right). \tag{6}$$

To simplify the writing, we use $h_{\mathcal{G}}(\hat{\mathbf{c}}, \hat{y})$ to denote the process of identifying (Section 4.2) and rectifying (Section 4.3) in AGAIN. We extend this definition to assess the accuracy of explanations generated by AGAIN:

$$\mathcal{A}^{\text{AGAIN}} := \prod_{c \in \mathbf{c}} \mathbb{P}_\delta \left( h_{\mathcal{G}}^{(m)} (\hat{\mathbf{c}}, \hat{y}) = c \right), \tag{7}$$

where $h_{\mathcal{G}}^{(m)} (\cdot)$ denotes the output of the factor graph reasoning with respect to the $m$-th concept.

**Lemma 1.** Given a factor graph $\mathcal{G}$, the following equation is valid:

$$A^{\text{AGAIN}} = \prod_{c \in \mathbf{c}} \mathbb{P}_\delta \left( \Delta_N (v_{\hat{c}}) > 0 \,|\, c \right), \tag{8}$$

where

$$\Delta_{\mathcal{N}} (v_{\hat{c}}) = \sum_{i \in |\mathcal{N}(v_{\hat{c}})|} (2w_i \psi_i - w_i). \tag{9}$$

**Proof.** First, $\mathcal{A}^{\text{AGAIN}} := \prod_{c \in \mathbf{c}} \mathbb{P}_\delta \left( h_{\mathcal{G}}^{(m)} (\hat{\mathbf{c}}, \hat{y}) = c \right)$ is known. Then, based on the properties of $\mathcal{G}$, $\mathbb{P}_\delta \left( h_{\mathcal{G}}^{(m)} (\hat{\mathbf{c}}, \hat{y}) = c \right)$ can be expressed equivalently as $\mathbb{P}_\delta \left( \mathbb{P} \left( \mathcal{F}_{v_{\hat{c}}} | v_{\hat{c}} = c \right) > \mathbb{P} \left( \mathcal{F}_{v_{\hat{c}}} | v_{\hat{c}} \neq c \right) | c \right)$. According to Eq. (1), we extend $\mathbb{P} \left( \mathcal{N} (v_{\hat{c}}) | v_{\hat{c}} = c \right) > \left( \mathcal{N} (v_{\hat{c}}) | v_{\hat{c}} \neq c \right)$ to a formulation that incorporates potential functions within $\mathcal{G}$:

$$\mathbb{P} \left( \mathcal{N} (v_{\hat{c}}) | v_{\hat{c}} = c \right) > \left( \mathcal{N} (v_{\hat{c}}) | v_{\hat{c}} \neq c \right)$$
$$\Rightarrow \sum_{i \in |\mathcal{N}(v_{\hat{c}})|} w_i \psi_i - \left( \sum_{i \in |\mathcal{N}(v_{\hat{c}})|} w_i (1 - \psi_i) \right) > 0$$
$$\Rightarrow \sum_{i \in |\mathcal{N}(v_{\hat{c}})|} w_i \psi_i - \sum_{i \in |\mathcal{N}(v_{\hat{c}})|} w_i + \sum_{i \in |\mathcal{N}(v_{\hat{c}})|} w_i \psi_i > 0$$
$$\Rightarrow \sum_{i \in |\mathcal{N}(v_{\hat{c}})|} 2 w_i \psi_i - w_i > 0,$$

then, we obtain:

$$\mathcal{A}^{\text{AGAIN}} = \prod_{c \in \mathbf{c}} \mathbb{P}_\delta \left( h_G^{(m)} (\hat{\mathbf{c}}, \hat{y}) = c \right) = \prod_{c \in \mathbf{c}} \mathbb{P}_\delta \left( (\mathcal{F}_{v_{\hat{c}}} | v_{\hat{c}} = c) > (\mathcal{F}_{v_{\hat{c}}} | v_{\hat{c}} \neq c) | c \right)$$
$$= \prod_{c \in \mathbf{c}} \mathbb{P}_\delta \left( \sum_{i \in |\mathcal{N}(v_{\hat{c}})|} 2 w_i \psi_i - w_i > 0 | c \right) = \prod_{c \in \mathbf{c}} \mathbb{P}_\delta \left( \sum_{i \in |\mathcal{N}(v_{\hat{c}})|} 2 w_i \psi_i - w_i > 0 | c \right)$$

$\square$

We observe that $\Delta_{\mathcal{N}} (v_{\hat{c}})$ determines the lower bound of Eq. (8). Further, we characterize the constraints on the concepts within $\mathcal{G}$ to impose bounds on the $\Delta_{\mathcal{N}} (v_{\hat{c}})$, thereby restricting its left-tailed probability below 0. To this end, considering a concept variable $v_{\hat{c}}$, we characterize the four cases of constraints imposed by $\mathcal{G}$ on $v_{\hat{c}}$ during reasoning with its neighbor factors $f_i \in \mathcal{N} (v_{\hat{c}})$:

$$L \leq \mathbb{P}_\delta (\psi_i | c) \leq U, \tag{10}$$

$$\begin{aligned} L_P^T \leq T^P &= \mathbb{P}_\delta (\psi_i = 1 | v_{\hat{c}} = c) \leq U_P^T \\ L_N^T \leq T^N &= \mathbb{P}_\delta (\psi_i = 0 | v_{\hat{c}} = 1 - c) \leq U_N^T \\ L_N^F \leq F^N &= \mathbb{P}_\delta (\psi_i = 0 | v_{\hat{c}} = c) \leq U_N^F \\ L_P^F \leq F^P &= \mathbb{P}_\delta (\psi_i = 1 | v_{\hat{c}} = 1 - c) \leq U_P^F. \end{aligned} \tag{11}$$

Moreover, we employ factor graph characterizations to represent the lower bound of $\Delta_{\mathcal{N}} (v_{\hat{c}})$. To this end, a lemma is introduced below (Gürel et al., 2021). This lemma illustrates an inequality property between $\Delta_{\mathcal{N}} (v_{\hat{c}})$ and factor graph characterizations.

**Lemma 2.** Suppose each factor has the optimal weight. Then there exists

$$\mathbb{E} (\Delta_{\mathcal{N}} (v_{\hat{c}}) | c) \geq Z_1 + Z_2 - \log \frac{c^{(1-2c)}}{1 - c}, \tag{12}$$

where

$$Z_1 = c \left( T^P \log \frac{L_P^T}{1 - U_P^F} + \left( 1 - T^P \right) \log \frac{1 - U_P^T}{1 - L_P^F} - F^N \log \frac{U_N^T}{L_N^F} + \left( 1 - F^N \right) \log \frac{1 - L_N^T}{1 - U_N^F} \right),$$
$$Z_2 = (1 - c) \left( T^N \log \frac{L_N^T}{U_N^F} + \left( 1 - T^N \right) \log \frac{1 - U_N^T}{1 - L_N^F} - F^P \log \frac{U_P^T}{L_P^F} + \left( 1 - F^P \right) \log \frac{1 - L_P^T}{1 - U_P^F} \right). \tag{13}$$

To facilitate the writing, we have opted for a simplified symbolic representation:

$$L^{\Delta_{\mathcal{N}(v_{\hat{c}}), c}} = Z_1 + Z_2 - \log \frac{c^{(1-2c)}}{1 - c}. \tag{14}$$

To further solve the bound of $\mathcal{A}^{\text{AGAIN}}$, we introduce another lemma (Chen et al., 2024). This lemma outlines the definition and properties of Hoeffding's inequality.

**Lemma 3.** Suppose the given $x_1, ..., x_t$ are independent random variables, and $x_t - x_{t-1} \in [a_t, b_t]$. The empirical mean of these variables denoted as $\bar{x} = (x_1 + ... + x_t)/t$. Then for any $\beta > 0$ there exists the Hoeffding's inequality as shown below:

$$\mathbb{P}\left(|\bar{x} - \mathbb{E}\left(\bar{x}\right)| \geq \beta\right) \leq 2\exp\left(-\frac{2\beta^2}{\sum_{i \in t}\left(\tau_i\right)^2}\right), \tag{15}$$

where $(b_i - a_i) \leq \tau_i$ and $\mathbb{E}(\cdot)$ denotes the expectation.

Herein, we provide the lower bound with respect to $\mathcal{A}^{\text{AGAIN}}$. This lower bound imposes a minimum accuracy constraint on AGAIN.

**Theorem 1.** For the AGAIN with $\mathcal{G}$, under the assumption that factors satisfying $Z_1 > 0$ and $Z_2 > 0$ are considered, the following inequality is established:

$$\mathcal{A}^{\text{AGAIN}} \geq \prod_{c \in \mathbf{c}}\left(1 - \mathbb{E}\left(\exp\left(\partial\right)\right)\right), \tag{16}$$

where

$$\partial = -2\frac{\left(L^{\Delta_{N(v_{\hat{c}}),c}}\right)^2}{\sum_{i \in \mathcal{N}}\left(\log\frac{U^T(1-L^F)}{L^T(1-U^F)}\right)^2}. \tag{17}$$

**Proof.** According to the symmetry of $\mathcal{G}$, we can derive the following equation:

$$\mathcal{A}^{\text{AGAIN}} = \prod_{c \in \mathbf{c}}\mathbb{P}_\delta\left(h_{\mathcal{G}}^{(c)}\left(\hat{\mathbf{c}}, \hat{y}\right) = c\right) = \prod_{c \in \mathbf{c}}\left(1 - \mathbb{P}_\delta\left(\Delta_{\mathcal{N}(v_{\hat{c}})} < 0 \,|c\right)\right).$$

According to Eq. (12), we can get

$$\begin{aligned}
\mathbb{P}_\delta\left(\Delta_{\mathcal{N}}\left(v_{\hat{c}}\right) < 0 \,|c\right) &= \mathbb{P}_\delta\left(\Delta_{\mathcal{N}(v_{\hat{c}})} - \mathbb{E}\left[\Delta_{\mathcal{N}(v_{\hat{c}})}\right] + \mathbb{E}\left[\Delta_{\mathcal{N}(v_{\hat{c}})}\right] < 0 \,|c\right) \\
&= \mathbb{P}_\delta\left(\left(\Delta_{\mathcal{N}(v_{\hat{c}})} - \mathbb{E}\left[\Delta_{\mathcal{N}(v_{\hat{c}})}\right]\right) < -\mathbb{E}\left[\Delta_{\mathcal{N}(v_{\hat{c}})}\right] \,|c\right) \\
&= \mathbb{P}_\delta\left(\left(\Delta_{\mathcal{N}(v_{\hat{c}})} - \mathbb{E}\left[\Delta_{\mathcal{N}(v_{\hat{c}})} \,|c\right]\right) < -\mathbb{E}\left[\Delta_{\mathcal{N}(v_{\hat{c}})} \,|c\right] \,|c\right) \\
&\leq \mathbb{P}_\delta\left(\left(\Delta_{\mathcal{N}(v_{\hat{c}})} - \mathbb{E}\left[\Delta_{\mathcal{N}(v_{\hat{c}})} \,|c\right]\right) < -L^{\Delta_{N(v_{\hat{c}}),c}} \,|c\right).
\end{aligned}$$

Substituting $\mathbb{P}_\delta\left(\Delta_{\mathcal{N}(v_{\hat{c}})} < 0 \,|c\right) \leq \mathbb{P}_\delta\left(\left(\Delta_{N(v_{\hat{c}})} - \mathbb{E}\left[\Delta_{\mathcal{N}(v_{\hat{c}})} \,|c\right]\right) < -L^{\Delta_{\mathcal{N}(v_{\hat{c}}),c}} \,|c\right)$ into Hoeffding's inequality and letting $\beta = L^{\Delta_{\mathcal{N}(v_{\hat{c}}),c}}$ in Lemma 3, we evidently derive the following inequality:

$$\mathbb{P}_\delta\left(\Delta_{\mathcal{N}(v_{\hat{c}})} < 0 \,|c\right) \leq \mathbb{P}_\delta\left(\Delta_{\mathcal{N}(v_{\hat{c}})} - \mathbb{E}\left[\Delta_{\mathcal{N}(v_{\hat{c}})} \,|c\right] < -L^{\Delta_{\mathcal{N}(v_{\hat{c}}),c}} \,|c\right) \leq \exp\left(-\frac{2\left(L^{\Delta_{\mathcal{N}(v_{\hat{c}}),c}}\right)^2}{\sum_{i \in \mathcal{N}}\left(\tau_i\right)^2}\right).$$

According to Lemma 3, we can use Eq. (9) and Eq. (12) to establish Hoeffding's inequality with respect to $\mathcal{A}^{\text{AGAIN}}$. If we consider $\Delta_{\mathcal{N}(v_{\hat{c}})}$ as the independent random variable $x_t$ in Lemma 3, we assume

$$x_{i+1}^{(\hat{c})} - x_i^{(\hat{c})} = w_i \exp\left(\psi_i\right) \quad \text{s.t.} \quad i \in |\mathcal{N}\left(v_{\hat{c}}\right)|,$$

according to Eq. (10), Eq. (11), and Eq. (12), we can infer $a_i$ and $b_i$ for $\psi_i$ in two cases. If $\psi_i = 1$, then there exists

$$\log\frac{L^T}{U^F} \leq w_i \exp\left(\psi_i\right) = w_i e \leq \log\frac{U^T}{L^F},$$

if $\psi_i = 0$, then there exists

$$\log\frac{1 - U^T}{1 - L^F} \leq w_i \exp\left(\psi_i\right) = w_i \leq \log\frac{1 - L^T}{1 - U^F}.$$

Since $w_i > 0$, it is evident that $w_i e > w_i$. Therefore, for both cases above, there is a uniform range interval that can be inferred

$$\log\frac{1 - U^T}{1 - L^F} \leq w_i \exp\left(\psi_i\right) \leq \log\frac{U^T}{L^F},$$

further, it can be inferred that $w_i \exp\left(\psi_i\right) \leq \tau_i$. We know $(b_i - a_i) \leq \tau_i$ and $a_i \leq w_i \exp\left(\psi_i\right) \leq b_i$ so $x_t - x_{t-1} \in [a_t, b_t]$ holds.

$$\log\frac{U^T}{L^F} - \log\frac{1-U^T}{1-L^F} = \log\frac{U^T(1-L^F)}{L^F(1-U^T)} \leq \tau_i \quad \text{s.t.} \quad i \in |\mathcal{N}\left(v_{\hat{c}}\right)|,$$

according to

$$\mathbb{P}_\delta\left(\Delta_N\left(v_{\hat{c}}\right) < 0 \,|c\right) \leq \exp\left(-\frac{2\left(L^{\Delta_N\left(v_{\hat{c}}\right),c}\right)^2}{\sum_{i\in N}\left(\tau_i\right)^2}\right),$$

we can be inferred that

$$\prod_{c\in\mathbf{c}} \mathbb{P}_\delta\left(h_{\mathcal{G}}^{(c)}\left(\widehat{\mathbf{c}}, \hat{y}\right) = c\right) = \prod_{c\in\mathbf{c}} \mathbb{P}_\delta\left(\Delta_{\mathcal{N}(v_{\hat{c}})} > 0 \,|c\right) = \prod_{c\in\mathbf{c}}\left(1 - \mathbb{P}_\delta\left(\Delta_{\mathcal{N}(v_{\hat{c}})} < 0 \,|c\right)\right)$$

$$\geq \prod_{c\in\mathbf{c}}\left(1 - \mathbb{E}\left[\exp\left(-\frac{2\left(L^{\Delta_{\mathcal{N}(v_{\hat{c}}),c}}\right)^2}{\sum_{i\in N}\left(\tau_i\right)^2}\right)\right]\right)$$

$$= \prod_{c\in\mathbf{c}}\left(1 - \mathbb{E}\left[\exp\left(-\frac{2\left(L^{\Delta_{\mathcal{N}(v_{\hat{c}}),c}}\right)^2}{\sum_{i\in\mathcal{N}}\left(\log\frac{U^T(1-L^F)}{L^F(1-U^T)}\right)^2}\right)\right]\right)$$

$\square$

## B.2 Lower Bound of Accuracy Versus Number of Factors in the Factor Graph

After analyzing the lower bound of concept accuracy, we aim to demonstrate a positive correlation between this lower bound and the number of factors in the factor graph. In simpler terms, an increase in the number of factors is directly proportional to an enhancement in predictive accuracy. This implies that $\mathcal{G}$ contribute to the predictive performance of the model. To facilitate the analysis, we introduce a lemma below (Gürel et al., 2021). This lemma illustrates an unequal relationship of the accuracy lower bound under specific factor graph characterizations.

**Lemma 4.** Suppose that the upper and lower bounds of each factor graph characteristic are identical, then there exists

$$\prod_{c\in\mathbf{c}}\left(1 - \mathbb{E}\left[\exp\left(-\frac{2\left(L^{\Delta_{\mathcal{N}(v_{\hat{c}}),c}}\right)^2}{\sum_{i\in N}\left(\log\frac{U^T(1-L^F)}{L^F(1-U^T)}\right)^2}\right)\right]\right) \geq \prod_{c\in\mathbf{c}}\left(1 - \exp\left(-2N\left(\Theta^T - \Theta^F\right)\right)\right), \quad (18)$$

where

$$\Theta^T := U^T = L^T \quad \Theta^F := U^F = L^F. \quad (19)$$

Furthermore, we present a theorem that highlights a significant correlation between the lower bound of concept accuracy and the number of factors in $\mathcal{G}$.

**Theorem 2.** For the given $\mathcal{G}$ in AGAIN, if it satisfies the assumptions of Lemma 4 and each factor in $\mathcal{G}$ satisfies $\Theta^T > \Theta^F$, the lower bound on the concept predictive accuracy increases strictly monotonically as the number of factors $N$ of the factor graph increases. Moreover, the lower bound of concept predictive accuracy with factor graph is strictly larger than the lower bound without factor graph.

**Proof.** According to Eq. (18), if $\Theta^T > \Theta^F$, we can achieve

$$\mathcal{A}^{\text{AGAIN}} = \prod_{c\in\mathbf{c}} \mathbb{P}_\delta\left(h_{\mathcal{G}}^{(c)}\left(\hat{\mathbf{c}}, \hat{y}\right) = c\right) \geq \prod_{c\in\mathbf{c}}\left(1 - \exp\left(-2N\left(\Theta^T - \Theta^F\right)\right)\right).$$

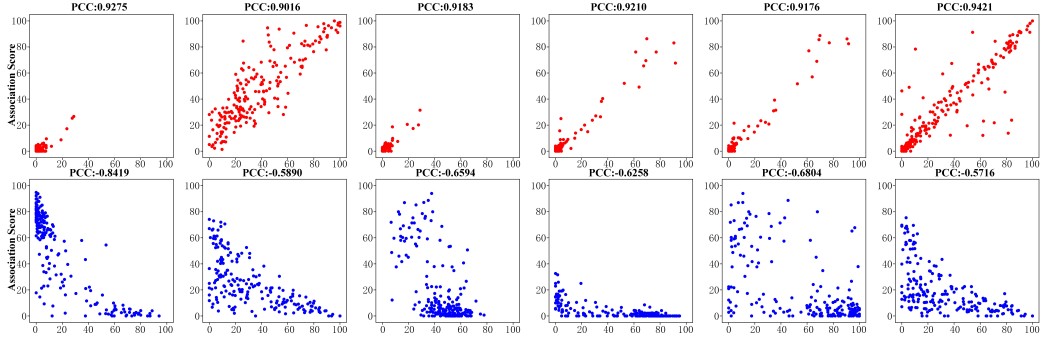

Figure 8: Correlation of association scores between concepts on the CUB dataset. The horizontal and vertical axes indicate the correlation scores of the two concepts. The top row shows the association scores for the 6 groups of concept pairs with PCCs > 0.8 (red scatter). The bottom row shows the association scores for the 6 sets of concept pairs with PCCs < -0.5 (blue scatter).

Let $\Theta^T - \Theta^F = \alpha > 0$ and $f(N) = 1 - \exp\left(-2N\left(\Theta^T - \Theta^F\right)\right)$. Then there exists the first order derivative $f'(N) = 2\alpha e^{-2\alpha N} > 0$ of $f(N)$. Therefore, $f(N)$ is monotonically increasing, i.e., $\mathcal{A}^{\text{AGAIN}}$ is a strictly monotonically increasing function with respect to $N$.

Notably, for the case without the factor graph (i.e., $N = 0$), we have $f(N) = 1 - \exp\left(-2N\left(\Theta^T - \Theta^F\right)\right) = 0$. And for a factor graph $\mathcal{G}$ with an arbitrary number of nodes, $f(N) > f(0) = 0$. Therefore, the concept predictive accuracy with the introduction of the factor graph is constantly greater than the case without the factor graph.

$\square$

## B.3 DISCUSSION

Our theoretical proof validates that $\mathcal{G}$ has the potential to enhance prediction accuracy, provided that the rules embedded in the factor graphs restrict concept activation in alignment with prior logic, satisfying the qualifying sufficient condition outlined in Theorem 2 (i.e., $T > F$). Intuitively, as the size of $N$ increases, the lower bound on $\mathcal{A}^{\text{AGAIN}}$ grows exponentially, leading to a substantial improvement in $\mathcal{A}^{\text{AGAIN}}$. We observe that Theorem 2 holds depending on the condition set by Lemma 4 (i.e., $U^T = L^T$ and $U^F = L^F$). Indeed, $U^T = L^T$ and $U^F = L^F$ are inherently met when the rules governing $\mathcal{G}$ adhere to prior logic. More specifically, if a factor satisfies $\Theta^T > \Theta^F$ and its neighboring variables adopt the same value as ground-truth labels, then its potential function value must be $e$. In this case, the conditional probability of the potential function is fixed.

## C EXPERIMENTAL DETAILS

### C.1 DATASETS AND DATA PREPROCESSING

All data processing and experiments are executed on a server with two Xeon-E5 processors, two RTX4000 GPUs and 64G memory. We construct logical rule sets on the three datasets respectively. The detailed information of data processing on both datasets is summarized as follows:

**Synthetic-MNIST.** The Synthetic-MNIST dataset is a composite dataset derived from the original MNIST dataset. Each category of the MNIST handwritten digits is treated as a concept, and four digits from different categories are concatenated to form a Synthetic-MNIST sample. Consequently, each Synthetic-MNIST sample contains 4 concept labels and a synthetic category label. The Synthetic-MNIST comprised 79,261 samples from 12 synthetic categories. The mapping of each synthetic category to concepts is shown in Table 9. According to Table 9, we construct the first-order logical rule set for Synthetic-MNIST. The samples, whose category label is 0, are taken as examples. We construct category-concept rules: $c_0 \Leftrightarrow y_0$, $c_2 \Leftrightarrow y_0$, $c_4 \Leftrightarrow y_0$, $c_6 \Leftrightarrow y_0$; concept-concept rules: $c_0 \oplus c_1$, $c_2 \oplus c_3$, $c_4 \oplus c_5$, $c_4 \oplus c_6$, $c_7 \oplus c_8$, $c_9 \oplus c_5$. Note that the logical rules used on the

Table 6: Statistics of rule and factor graphs constructed on three datasets.

| | Dataset | Synthetic -MNIST | CUB | MIMIC -III EWS |
|---|---|---|---|---|
| Logical Rule | Category-concept | 33 | 11,300 | 144 |
| | Concept-concept | 13 | 3763 | 35 |
| Factor Graph | Concept Variable | 10 | 112 | 22 |
| | Category Variable | 12 | 200 | 16 |
| | Logical Factor | 46 | 15,063 | 179 |

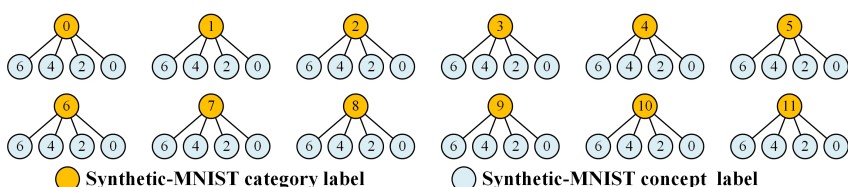

Figure 9: Category labels and concept labels for Synthetic-MNIST.

synthetic dataset can encompass the entire knowledge required for the downstream tasks. Therefore, we randomly omit a subset of these rules to simulate the incompleteness of explicit knowledge[2].

**CUB.** The CUB (Caltech-UCSD Birds-200-2011) dataset comprises 11,788 images of birds distributed across 200 categories. Each image is annotated with 312 high-level semantic labels, such as wing color and beak shape, in addition to a single category label. We have retained 112 crucial semantic labels as concepts following a denoising process. Furthermore, potential associations between concepts and categories are manually labeled by the bird experts and quantified as an association score in the interval $[0, 100]$. Association scores approaching 100 or 0 signify a significant degree of coexistence or exclusion, respectively, between the concept and the category. Convergence towards 50 indicates the absence of any discernible association. We formulate the logical rule set based on the association scores. Association scores near 100 are regarded as category-concept rules representing coexistence constraints, while association scores approaching 0 are identified as category-concept rules indicating exclusion constraints. To construct concept-concept rules, we create a score vector that includes association scores corresponding to a concept under each category. We calculate the Pearson correlation coefficient (PCC) between each score vector, an elevated PCC between score vectors suggests a greater similarity between the two concepts. As illustrated in Figure 8, we visualized the association scores of 6 sets of exclusive concepts and 6 sets of coexisting concepts across 200 categories, respectively. We observe that for concept groups with PCCs >0.8, their association scores are positively linearly correlated. While concept groups with the PCC <-0.5 showed negative linear correlation. Therefore, we construct coexistence rules for groups of concepts with PCCs > 0.8 and exclusion rules for groups of concepts with PCCs < -0.5[3].

**MIMIC-III EWS.** The MIMIC-III EWS dataset is a medical dataset for an early warning score (EWS) prediction task, comprising electronic health records from 17,289 patients. The EWS is the patient's vital sign score, ranging from 0 to 15. Deviation from normal vital signs results in an increase in EWS. We utilize 15 input attributes of patients to predict 16 labeled categories (each integer value of EWS as a category). Additionally, we predefine 22 concepts related to vital signs (such as body temperature, mean blood pressure, etc.) based on the Hard AR (Havasi et al., 2022) recommendations for generating explanations. We directly establish category-concept rules based on the existing probability analysis results from Hard AR and formulate concept-concept rules using cosine similarity[4].

The statistics of the rules and factor graphs constructed on the aforementioned dataset are presented in Table 6, illustrating the number of each type of logical rules and each type of nodes in $\mathcal{G}$.

---

[2]http://yann.lecun.com/exdb/mnist/
[3]http://www.vision.caltech.edu/visipedia/CUB-200.html
[4]https://physionet.org/content/mimiciii/1.4/

### C.2 CONCEPT-LEVEL INTERPRETABLE MODEL BASELINES

To evaluate the comprehensibility of the explanations, we compared AGAIN with 8 popular concept-level interpretable model baselines. The baseline models are listed as follows:

- **CBM** (Koh et al., 2020a) is a classical interpretable neural network implemented with the concept bottleneck structure.
- **Hard AR** (Havasi et al., 2022) expands the concept set with side-channels, and predicts categories with binary concept vectors.
- **ICBM** (Chauhan et al., 2023) introduces interactive policy learning with cooperative prediction to filter important concepts.
- **PCBM** (Yuksekgonul et al., 2023) introduces a concept-level self-interpretation module that automatically captures concepts through multimodal models.
- **ProbCBM** (Kim et al., 2023) uses probabilistic concept embedding to model uncertainty in concept predictions and explains predictions in terms of the likelihood that the concept exists.
- **Label-free CBM** (Oikarinen et al., 2023) introduces a multimodal model to generate a predefined concept set for the inputs and learns the mapping between input features and the concept set.
- **ProtoCBM** (Huang et al., 2024) utilizes cross-layer alignment and cross-image alignment to learn the mapping of different parts of the feature map to concept predictions, thereby promoting the model to capture trustworthy concept prototypes.
- **ECBMs** (Xu et al., 2024) utilizes conditional probabilities to quantify predictions, concept corrections, and conditional dependencies to capture higher-order nonlinear interactions between concepts for improving the reliability of concept activations.

### C.3 EVALUATION METRIC

- **P-ACC**. P-ACC measures the validity of concept-level explanations by reapplying rectified explanations to the category predictor and examining the outcomes of category predictions.
- **E-ACC**. E-ACC measures the similarity between a concept-level explanation and the ground-truth explanatory concept set, which is computed as follows:

$$E\text{-}ACC = \mathbb{E}\left[\frac{\sum_{m \in M} \mathbb{I}\left[\hat{c}_{re,m} = c_m\right]}{M}\right], \tag{20}$$

where $\hat{c}_{re,m}$ represents the binary transformation result for the $m$-th concept activation, and $c_m$ denotes the ground-truth label of the $m$-th concept.

- **LSM**. Since the comprehensibility of explanations is based on prior human logics, the extent to which an explanation adheres to logical rules can be utilized as a criterion for evaluating its comprehensibility. In this experiment, we formulated LSM to quantify the extent to which an explanation adheres to the prior human logics rules. This metric is employed to evaluate the comprehensibility of the explanation. A higher LSM indicates a higher comprehensibility of the explanation. In particular, we calculate the LSM by considering the weighted sum of potential functions within the factor graph. For an explanation **a**, the definition of LSM is as follows:

$$LSM = \mathbb{E}\left[\frac{\exp\left(\sum_{i \in N} w_i \psi_i\right)}{\prod_{i \in N} w_i e}\right]. \tag{21}$$

The maximum value for LSM is 1, signifying that the explanation adheres to all the logical rules encoded in $\mathcal{G}$.

- **IR and SR**. In this experiment, IR and SR are devised to evaluate the capacity of the factor graph for recognizing perturbations. IR quantifies the rate at which perturbed instances are identified by the factor graph, computed as $\mathbb{E}\left[\mathbb{P}\left(\mathcal{V}^c \mid \mathcal{V}^y\right) > \partial \cdot \vee \mathbb{P}\left(\mathcal{V}^c \mid \mathcal{V}^y\right) \mid x + \delta\right]$, while SR measures the rate at which benign instances are allowed to pass through the factor graph, computed as $\mathbb{E}\left[\mathbb{P}\left(\mathcal{V}^c \mid \mathcal{V}^y\right) > \partial \cdot \vee \mathbb{P}\left(\mathcal{V}^c \mid \mathcal{V}^y\right) \mid x\right]$.

Remarkably, only CBM, ProtoCBM, ECBMs, and ProbCBM baselines are suitable to Synthetic-MNIST dataset, as the concepts in this dataset is synthetic and cannot be automatically captured by other baselines.

### C.4 IMPLEMENTATION DETAILS

AGAIN is implemented in PyTorch 1.1.0 based on Python 3.7.13. We construct $\mathcal{G}$ by instantiating the $\mathcal{G}$ as a Markov logic network in Pracmln 1.2.4. Within the AGAIN, a fully connected layer is utilized as the category predictor. InceptionV3 (Koh et al., 2020a), a popular convolutional neural network structure, is employed as the concept predictor trained on real-world datasets. We use a convolutional neural network with two convolutional layers and normalization operations as the concept predictor trained on Synthetic-MNIST. We train the concept predictor (real-world datasets) for 500 epochs, the concept predictor (Synthetic-MNIST) for 30 epochs, and the category predictor for 15 epochs. We leverage the sgd optimizer with a learning rate of 0.01 to optimize the model. We to mitigate the overfitting, weight decay of 0.00004 was configured. In the experiment, $\partial$ is set to 0.9. All experiments were repeated 4 times and the average of the results is reported.

### C.5 ESTIMATION OF WEIGHTS

Due to the differences in the datasets, we use two methods to set rule weights in the factor graphs: prior setting and likelihood estimation. For the CUB dataset, we use the prior-based weighting method, as it provides predefined confidence for the mapping relations between concepts and categories. Therefore, we treat the weights as hyperparameters and directly convert the confidence into corresponding weights.

For the MIMIC-III EWS and Synthetic-MNIST datasets, we use likelihood estimation to learn the weights, as these datasets do not contain confidence information. Specifically, we directly apply standard maximum likelihood estimation Yang et al. (2022) to learn the weights of each factor. We assume that the samples in the training set should satisfy all logical rules when there is no perturbation. Therefore, after assigning the concept activations of the samples to the factor graph, the weights of the factors should maximize the conditional probability $\mathbb{P}\left(\mathcal{V}^c \mid \mathcal{V}^y\right)$. In summary, we minimize the negative log-likelihood function:

$$\mathbf{w} = \arg\min_{\mathbf{w}} \left\{ -\sum_N \log\left(\mathbb{P}\left(\mathcal{V}^c \mid \mathcal{V}^y, \mathbf{w}\right)\right) \right\}, \tag{22}$$

where $N$ denotes the number of samples in the training set, and $\mathbf{w}$ denotes the weight vector formed by concatenating the weights of all factors in the factor graph.

## D MORE EXPERIMENTAL RESULTS

### D.1 ABLATION ANALYSIS OF FACTOR GRAPHS

To validate the necessity of using factor graphs, we conducted a set of ablation studies comparing the impact on LSM with and without factor graphs across three datasets. The altered model without factor graphs is denoted as "w/o Factor Graph," which only uses a predefined set of logic rules, and the model with factor graphs is denoted as "w/ Factor Graph," which encodes the logic rules using a factor graph. The experimental results, shown in Table 7, indicate that the model using factor graphs consistently yields better LSMs across all three datasets compared to the model that uses only the logic rule set. This improvement is attributed to the uncertainty reasoning capability of factor graphs, which can estimate the confidence levels of different rules and assign appropriate weights accordingly. Notably, on the Synthetic-MNIST dataset, the LSMs achieved with just the logic rule set are comparable to those with factor graphs. This is because most logic rules in the Synthetic-MNIST dataset are synthesized and deterministic by nature. Purely deterministic logic rules are often less applicable in real-world scenarios. Therefore, factor graphs offer an irreplaceable advantage in detecting erroneous explanations in real-world scenarios.

Table 7: Comparison of LSM with and without the factor graph.

| Dataset | Methods | $\delta_k$ | | | | $\delta_u$ | | | |
|---------|---------|------------|------------|------------|------------|------------|------------|------------|------------|
| | | $\epsilon$=4 | $\epsilon$=8 | $\epsilon$=16 | $\epsilon$=32 | $\epsilon$=4 | $\epsilon$=8 | $\epsilon$=16 | $\epsilon$=32 |
| CUB | w/o Factor Graph | 89.0(5.2) | 87.6(6.7) | 87.1(7.1) | 86.1(6.2) | 89.0(5.2) | 87.6(6.7) | 87.1(7.1) | 86.1(6.2) |
| | w/ Factor Graph | **92.4(1.2)** | **93.1(2.3)** | **93.8(1.9)** | **91.5(1.7)** | **94.5(1.6)** | **93.3(1.7)** | **93.8(1.4)** | **92.1(2.1)** |
| MIMIC -III EWS | w/o Factor Graph | 89.9(4.1) | 89.8(7.2) | 88.9(7.6) | 87.4(6.9) | 89.9(4.1) | 89.8(7.2) | 88.9(7.6) | 87.4(6.9) |
| | w/ Factor Graph | **96.1(0.7)** | **94.2(1.4)** | **96.1(1.2)** | **94.2(1.2)** | **94.0(2.7)** | **94.1(6.3)** | **94.2(2.4)** | **92.3(4.7)** |
| Synthetic -MNIST | w/o Factor Graph | 98.1(0.8) | 97.3(1.5) | 96.9(4.9) | 94.9(3.8) | 98.1(0.8) | 97.3(1.5) | 96.9(4.9) | 94.9(3.8) |
| | w/ Factor Graph | **98.2(1.4)** | **97.9(1.3)** | **97.8(1.4)** | **95.6(1.2)** | **98.2(1.4)** | **97.9(1.3)** | **97.8(1.6)** | **97.8(2.0)** |

## D.2 COMPUTATIONAL EFFICIENCY ANALYSIS

We assess the potential impact of increasing factor graph size on computational efficiency. Specifically, we evaluate four factor graph sizes, containing 30, 60, 90, and 112 concepts, on the CUB dataset, measuring their running time and LSM performance, respectively. To the best of our knowledge, CUB is currently the dataset with the highest number of concepts (with 112 concept labels). In addition, it is not necessary to discuss other datasets that have more concepts than CUB. Too many concepts can lead to lengthy explanations, reducing comprehensibility. The evaluation is performed under the unknown perturbation of $\epsilon$=32.

We report the running time required to detect and correct erroneous explanations and the LSM of the corrected explanations for all 4 scales of factor graphs (see Figure 10). According to the results, the running time of the factor graph is inevitably higher than that of the other baselines due to the extra steps of detection and correction required for the factor graph. However, this extra overhead can be contained to the millisecond level. Furthermore, the running time is indeed proportional to the size of the factor graph, but there is no exponential explosion, suggesting that AGAIN has good scalability.

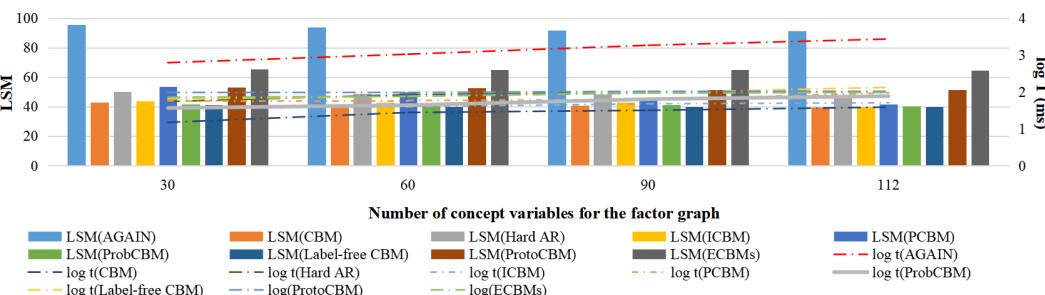

Figure 10: The computational efficiency analysis of factor graphs with different sizes.

## D.3 ANALYSIS OF INTERVENTION STRATEGIES

We compare the computational overhead associated with different intervention strategies for correcting concept activations. Specifically, we use the greedy strategy and the heuristic strategy as baselines for comparison. For the heuristic strategy, we provide the indexes of incorrect concepts, while for the concept intervention strategy, we supply the indexes of factors with a potential function of 0. We evaluate the computational overhead of these three strategies on the CUB dataset under a perturbation of $\epsilon$=32. 5 samples are selected and randomly modified between 1 and 10 concepts in each sample. Figure 11 illustrates the average number of interventions performed by different strategies across these samples. The experimental results indicate that the number of interventions increases for all three strategies as the number of incorrect concepts rises. When the number of incorrect concepts is fewer than 10, the differences in the number of interventions among the three strategies are minimal. Notably, the number of perturbed wrong concepts is typically small. For instance, in the CUB dataset, the number of wrong concepts is typically only 1 to 10.

In addition, we theoretically analyze the computational complexity of the intervention strategy of AGAIN. $\{\vee, \wedge, \neg\}$ is proved to be sufficient to express all logical relations. Based on this, if two concepts are randomly selected from a set of $M$ concepts, there are at most $\frac{M(M-1)}{2}$ possible combinations. The maximum number of distinct logical rules between two concepts is 8, specifically:

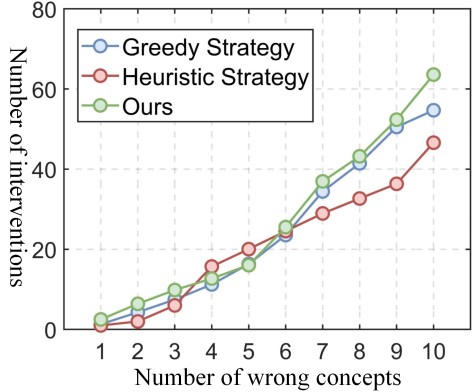

Figure 11: Comparison of intervention numbers for different intervention strategies.

Table 8: Comparisons of LSM for our AGAIN on the Synthetic-MNIST dataset.

| Method | clear | $\delta_k$ | | $\delta_u$ | |
| --- | --- | --- | --- | --- | --- |
| | | $\epsilon$=4 | $\epsilon$=8 | $\epsilon$=4 | $\epsilon$=8 |
| CBM | 97.8(0.6) | 92.6(3.5) | 86.8(4.6) | 92.6(4.7) | 86.3(5.4) |
| ProbCBM | 98.6(1.1) | 92.9(4.2) | 87.3(3.4) | 92.4(3.2) | 86.7(3.1) |
| ProtoCBM | 97.3(0.2) | 94.5(0.8) | 90.7(1.2) | 94.7(2.7) | 88.9(4.3) |
| ECBMs | 97.5(0.7) | 92.6(3.5) | 88.3(2.8) | 92.6(3.4) | 88.7(2.7) |
| CBM-AT | 93.5(0.8) | 93.0(2.4) | 90.4(3.1) | 92.6(2.4) | 86.3(1.3) |
| ProbCBM-AT | 93.5(0.7) | 92.9(2.7) | 90.4(2.6) | 92.4(3.4) | 86.7(2.1) |
| ProtoCBM-AT | 96.1(1.7) | 92.1(1.9) | 90.7(1.0) | 87.5(1.4) | 84.3(3.1) |
| ECBMs-AT | 95.9(1.4) | 90.4(4.2) | 89.5(1.7) | 88.0(1.2) | 84.7(2.5) |
| LEN | 98.6(0.3) | 97.6(1.4) | 95.5(2.9) | 98.6(1.4) | 95.5(2.9) |
| DKAA | 98.4(1.0) | 98.1(0.8) | 96.7(1.8) | 98.1(0.8) | 96.7(1.8) |
| MORDAA | 97.9(0.8) | 98.5(0.9) | 95.8(1.6) | 98.5(0.9) | 95.8(1.6) |
| DeepProblog | 97.5(1.4) | 97.6(3.5) | 95.6(3.5) | 98.6(3.5) | 95.6(3.5) |
| MBM | 98.7(0.1) | 97.7(0.7) | 95.1(1.2) | 97.7(0.7) | 95.1(1.2) |
| C-HMCNN | 98.1(0.4) | 96.8(1.2) | 94.5(1.6) | 96.8(1.2) | 94.5(1.6) |
| AGAIN (Ours) | **98.9(1.3)** | **97.8(1.4)** | **95.6(1.2)** | **97.8(1.6)** | **97.8(2.0)** |

$A \wedge B$; $\neg A \wedge B$; $A \wedge \neg B$; $\neg A \wedge \neg B$; $A \vee B$; $\neg A \vee B$; $A \vee \neg B$; $\neg A \vee \neg B$ Thus, the maximum number of factors is $4M(M-1)$. With a maximum of 3 interventions per rule (intervene A, intervene B, and both), the total number of interventions is $12M(M-1)$, resulting in a complexity of $O(M^2)$.

## D.4 EXPERIMENTAL RESULTS ON SYNTHETIC-MNIST

**Comprehensibility of Explanations.** Table 8 illustrates the LSM results of our proposed AGAIN in comparison with 10 baselines on the Synthetic-MNIST dataset. The results indicate that the explanations produced by AGAIN for the synthetic dataset are equally comprehensible, implying that the performance of AGAIN exhibits generalizability. Specifically, when the unknown perturbation magnitude is 8, the LSM of AGAIN exhibits an average increase of 41.83% compared to the baseline with attributional training. Furthermore, in benign environments, the LSM of AGAIN outperforms other baselines due to its ability to maintain optimal model parameters without adjusting them to accommodate the effects of perturbations.

**Rectification of Interactive Intervention Switch.** Figure 7(c) illustrates the visual representations of the rectified explanations generated by AGAIN. The activated handwritten digit concepts in the explanations align seamlessly with the semantics of synthetic images, confirming the logical completeness and semantic richness of the rectified explanations. This outcome validates the continued effectiveness of interactive intervention switch strategy on the synthetic dataset.

## D.5 E-ACC AND P-ACC ON EACC-MIMIC-IIIEWS AND SYNTHETIC-MNIST

We present the E-ACC of AGAIN on EACC-MIMIC-IIIEWS and Synthetic-MNIST in comparison to all baseline methods, as shown in Tables 9, and 10, respectively. The experimental results show that AGAIN is optimal for E-ACC on two datasets.

Secondly, we report the P-ACC of AGAIN on the two datasets, as shown in Tables 11, and 12, respectively. Notably, since perturbations do not impact the final predictions, the P-ACC remains consistent across different levels of perturbation. Furthermore, the factor graph does not improve the predictive accuracy of the categories, making the P-ACC of AGAIN comparable to that of the other baselines. Notably, since the methods based on knowledge integration are not retrained. Therefore, there is no difference between the effects of known and unknown perturbations on these methods. Therefore, their results under known and unknown perturbations are the same.

Table 9: Comparisons of E-ACC between AGAIN and baselines on MIMIC-III EWS.

| Method | clear | $\delta_k$ | | | | $\delta_u$ | | | |
|---|---|---|---|---|---|---|---|---|---|
| | | $\epsilon=4$ | $\epsilon=8$ | $\epsilon=16$ | $\epsilon=32$ | $\epsilon=4$ | $\epsilon=8$ | $\epsilon=16$ | $\epsilon=32$ |
| CBM-AT | 97.1(1.2) | 92.8(1.1) | 90.3(1.3) | 87.2(3.1) | 85.4(1.7) | 90.5(1.4) | 87.7(1.1) | 84.1(2.6) | 76.8(2.1) |
| Hard AR-AT | 97.7(0.4) | 92.9(2.1) | 89.9(1.2) | 87.4(2.4) | 84.3(1.4) | 90.1(2.4) | 84.1(0.6) | 80.3(3.1) | 78.8(6.7) |
| ICBM-AT | 97.8(1.4) | 91.9(1.4) | 86.8(1.5) | 84.5(2.7) | 83.2(3.2) | 89.6(1.6) | 85.9(2.3) | 82.7(2.1) | 79.1(4.2) |
| PCBM-AT | 96.9(1.7) | 92.6(1.4) | 86.3(2.1) | 83.2(0.8) | 83.0(2.1) | 90.5(1.2) | 84.6(2.7) | 81.5(3.2) | 77.4(3.9) |
| ProbCBM-AT | 98.1(0.3) | 92.1(1.2) | 90.3(1.1) | 87.4(1.3) | 84.4(2.4) | 90.0(2.2) | 85.5(1.2) | 84.2(1.0) | 81.6(3.2) |
| Label-free CBM-AT | 97.1(1.6) | 93.1(2.1) | 90.3(1.5) | 87.5(3.2) | 84.6(2.3) | 91.0(2.5) | 86.7(1.7) | 83.7(2.2) | 80.6(4.1) |
| ProtoCBM-AT | 97.8(0.8) | 93.4(1.3) | 92.1(1.0) | 86.7(1.4) | 85.6(2.1) | 92.1(0.6) | 89.5(1.2) | 85.8(0.9) | 81.2(2.4) |
| ECBMs-AT | **98.0(1.2)** | **93.6(1.7)** | 92.7(2.6) | 88.3(1.7) | 88.1(2.2) | 90.4(1.8) | 86.4(2.2) | 83.4(1.8) | 80.2(3.1) |
| LEN | 97.7(0.2) | 92.2(2.4) | 91.6(1.2) | 86.7(2.6) | 83.2(0.9) | 92.2(2.4) | 91.6(1.2) | 86.7(2.6) | 83.2(0.9) |
| DKAA | 96.5(1.3) | 91.7(1.3) | 92.9(0.8) | 87.5(1.4) | 85.1(3.2) | 91.7(1.3) | 92.9(0.8) | 87.5(1.4) | 85.1(3.2) |
| MORDAA | 97.1(1.4) | 91.5(3.7) | 90.5(3.1) | 84.3(1.4) | 81.5(2.1) | 91.5(3.7) | 90.5(3.1) | 84.3(1.4) | 81.5(2.1) |
| DeepProblog | 96.3(1.2) | 90.7(1.0) | 89.7(3.1) | 85.2(1.3) | 81.7(1.3) | 90.7(1.0) | 89.7(3.1) | 85.2(1.3) | 81.7(1.3) |
| MBM | 97.9(1.0) | 91.9(2.2) | 90.1(0.6) | 86.7(0.9) | 82.6(1.3) | 91.9(2.2) | 90.1(0.6) | 86.7(0.9) | 82.6(1.3) |
| C-HMCNN | 97.6(0.7) | 93.4(1.1) | 91.0(3.1) | 87.4(2.4) | 84.9(2.8) | 93.4(1.1) | 91.0(3.1) | 87.4(2.4) | 84.9(2.8) |
| AGAIN | 97.5(0.1) | 93.0(1.3) | **93.2(1.3)** | **93.0(1.4)** | **93.0(1.2)** | **93.0(1.7)** | **93.6(1.6)** | **93.3(1.9)** | **93.2(1.2)** |

Table 10: Comparisons of E-ACC between AGAIN and baselines on Synthetic-MNIST.

| Method | clear | $\delta_k$ | | | | $\delta_u$ | | | |
|---|---|---|---|---|---|---|---|---|---|
| | | $\epsilon=4$ | $\epsilon=8$ | $\epsilon=16$ | $\epsilon=32$ | $\epsilon=4$ | $\epsilon=8$ | $\epsilon=16$ | $\epsilon=32$ |
| CBM-AT | 98.4(0.1) | 98.1(1.2) | 96.1(1.4) | 95.2(1.5) | 90.3(1.6) | 97.1(1.2) | 96.5(2.1) | 92.1(1.5) | 89.4(2.1) |
| Hard AR-AT | 97.9(0.1) | 97.6(0.5) | 96.2(0.2) | 93.7(1.2) | 91.4(1.1) | 95.6(0.5) | 93.1(1.2) | 91.7(1.0) | 87.9(2.1) |
| ICBM-AT | 98.2(0.3) | 98.2(0.2) | 95.1(0.6) | 92.1(1.5) | 90.5(1.4) | 95.1(1.1) | 94.7(1.1) | 92.1(1.5) | 88.2(1.4) |
| PCBM-AT | 97.9(0.2) | 97.4(1.1) | 96.1(0.4) | 95.5(1.0) | 92.3(1.3) | 95.2(1.3) | 94.1(1.2) | 91.0(1.4) | 90.1(1.2) |
| ProbCBM-AT | 98.1(0.2) | 97.9(0.2) | 95.1(1.0) | 93.1(1.4) | 90.5(1.3) | 94.3(1.1) | 93.1(1.2) | 91.0(0.8) | 88.2(1.1) |
| Label-free CBM-AT | 97.5(0.7) | 97.7(0.6) | 95.9(1.2) | 92.5(1.3) | 91.6(1.3) | 95.1(1.8) | 92.9(3.1) | 90.2(4.2) | 87.2(2.3) |
| ProtoCBM-AT | **98.8(0.3)** | **98.5(1.2)** | 97.1(1.2) | 95.6(2.1) | 93.9(2.1) | 94.1(1.0) | 92.5(2.1) | 90.2(3.2) | 89.3(2.4) |
| ECBMs-AT | 98.5(0.6) | 98.1(0.6) | 97.1(1.2) | 95.3(1.3) | 92.1(2.5) | 95.2(1.1) | 92.5(1.7) | 90.1(1.3) | 88.6(1.6) |
| LEN | 98.4(1.1) | 98.4(1.2) | 96.9(1.4) | 95.4(0.9) | 92.8(1.2) | 98.4(1.2) | 96.9(1.4) | 95.4(0.9) | 92.8(1.2) |
| DKAA | 98.6(0.7) | 98.4(0.9) | 97.2(1.4) | 94.1(1.2) | 92.5(2.1) | 98.4(0.9) | 97.2(1.4) | 94.1(1.2) | 92.5(2.1) |
| MORDAA | 98.3(0.7) | 97.9(1.4) | 95.2(1.2) | 92.1(1.4) | 90.6(2.6) | 97.9(1.2) | 95.2(1.2) | 92.1(1.4) | 90.6(2.6) |
| DeepProblog | 97.5(1.1) | 97.3(2.1) | 95.3(2.4) | 92.7(2.1) | 89.4(3.2) | 97.3(2.1) | 95.3(2.4) | 92.7(2.1) | 89.4(3.2) |
| MBM | 98.7(1.3) | 97.4(1.5) | 96.4(1.3) | 91.6(1.3) | 89.4(2.2) | 97.4(1.5) | 96.4(1.3) | 91.6(1.3) | 89.4(2.2) |
| C-HMCNN | 98.8(0.9) | 96.7(2.1) | 95.1(2.4) | 93.2(2.5) | 90.5(2.5) | 96.7(2.1) | 95.1(2.4) | 93.2(2.5) | 90.5(2.5) |
| AGAIN | 98.1(0.5) | 98.1(0.5) | **97.3(1.3)** | **97.3(2.9)** | **97.2(2.1)** | **98.2(0.9)** | **97.4(1.0)** | **97.1(1.2)** | **96.8(1.4)** |

Table 11: Comparisons of P-ACC between AGAIN and baselines on MIMIC-III EWS.

| Method | clear | $\delta_k$ | | | | $\delta_u$ | | | |
|---|---|---|---|---|---|---|---|---|---|
| | | $\epsilon=4$ | $\epsilon=8$ | $\epsilon=16$ | $\epsilon=32$ | $\epsilon=4$ | $\epsilon=8$ | $\epsilon=16$ | $\epsilon=32$ |
| CBM-AT | 49.7(1.2) | 49.6(1.2) | 49.1(1.1) | 49.2(1.4) | 49.6(1.3) | 49.7(1.2) | 49.3(2.4) | 49.8(1.1) | 49.5(1.4) |
| Hard AR-AT | 48.5(2.1) | 49.1(1.5) | 49.1(1.2) | 49.5(1.1) | 49.6(1.2) | 49.1(1.5) | 49.1(2.3) | 48.9(1.1) | 49.0(1.6) |
| ICBM-AT | 49.1(1.6) | 49.6(1.4) | 49.1(1.4) | 49.5(1.5) | 49.4(1.1) | 48.9(1.1) | 49.4(1.4) | 49.5(1.7) | 49.9(1.4) |
| PCBM-AT | 49.6(1.4) | 49.9(1.3) | 49.8(2.1) | 50.1(1.3) | 49.8(1.2) | 49.7(1.2) | 49.1(1.3) | 49.3(1.4) | 50.1(1.1) |
| ProbCBM-AT | 49.8(0.8) | 49.8(1.0) | 51.4(2.1) | 50.2(1.3) | 49.6(1.5) | 50.2(1.1) | 49.4(1.3) | 49.6(1.1) | 49.7(0.9) |
| Label-free CBM-AT | 49.7(1.3) | 49.7(1.2) | 49.7(1.1) | 49.6(1.4) | 49.5(1.2) | 49.6(1.1) | 49.5(1.5) | 49.6(1.2) | 49.7(1.1) |
| ProtoCBM-AT | 49.6(1.2) | 49.5(1.1) | 48.9(2.0) | 49.1(1.4) | 49.3(1.3) | 49.3(1.2) | 48.9(2.1) | 49.0(1.3) | 49.3(1.2) |
| ECBMs-AT | 50.9(1.5) | 50.2(1.3) | 49.6(2.0) | 49.8(1.3) | 49.8(1.6) | 49.5(0.7) | 49.6(2.5) | 49.3(1.2) | 49.8(1.2) |
| LEN | 49.6(1.6) | 49.7(1.3) | 49.4(1.4) | 49.8(0.4) | 49.5(1.6) | 49.7(1.3) | 49.4(1.4) | 49.8(0.4) | 49.5(1.6) |
| DKAA | 49.4(1.1) | 49.8(0.1) | 50.5(0.2) | 49.7(1.2) | 49.8(1.2) | 49.8(0.1) | 50.5(0.2) | 49.7(1.2) | 49.8(1.2) |
| MORDAA | 49.5(0.1) | 49.8(1.2) | 49.7(1.3) | 50.1(1.4) | 49.4(1.6) | 49.8(1.2) | 49.7(1.3) | 50.1(1.4) | 49.4(1.6) |
| DeepProblog | 48.4(1.2) | 48.7(1.3) | 48.4(1.2) | 48.5(1.1) | 48.2(1.1) | 48.7(1.3) | 48.4(1.2) | 48.5(1.1) | 48.2(1.1) |
| MBM | 49.4(1.3) | 50.1(0.6) | 49.3(1.7) | 49.4(0.5) | 49.3(1.7) | 50.1(0.6) | 49.3(1.7) | 49.4(0.5) | 49.3(1.7) |
| C-HMCNN | 49.4(0.8) | 49.5(1.1) | 49.4(1.1) | 49.2(1.5) | 49.2(1.2) | 49.5(1.1) | 49.4(1.1) | 49.2(1.5) | 49.2(1.2) |
| AGAIN | 55.1(2.9) | 49.7(1.1) | 49.1(2.2) | 49.4(0.1) | 49.3(2.7) | 48.0(1.5) | 48.3(1.2) | 49.4(1.9) | 52.5(0.9) |

# E IMPLEMENTATION DETAILS OF ADVERSARIAL ATTACKS

In this section, we provide implementation details of adversarial attacks against concept-level explanations. Specifically, adversarial attacks against concept-level explanations can be categorized into three types: erasure attacks, introduction attacks, and confounding attacks.

Table 12: Comparisons of P-ACC between AGAIN and baselines on Synthetic-MNIST.

| Method | clear | $\delta_k$ | | | | $\delta_u$ | | | |
|---|---|---|---|---|---|---|---|---|---|
| | | $\epsilon$=4 | $\epsilon$=8 | $\epsilon$=16 | $\epsilon$=32 | $\epsilon$=4 | $\epsilon$=8 | $\epsilon$=16 | $\epsilon$=32 |
| CBM-AT | 98.2(0.1) | 98.2(0.1) | 98.1(0.4) | 98.5(0.2) | 98.1(0.5) | 98.5(1.2) | 98.5(0.6) | 97.6(0.3) | 98.2(0.5) |
| Hard AR-AT | 98.4(0.3) | 98.3(1.1) | 98.4(0.2) | 98.4(0.2) | 98.2(0.1) | 98.1(0.3) | 98.8(0.5) | 98.1(0.6) | 98.2(0.5) |
| ICBM-AT | 97.5(1.4) | 98.3(0.1) | 97.4(0.2) | 98.4(0.8) | 97.3(0.5) | 98.2(1.1) | 98.5(0.3) | 97.1(0.1) | 98.6(0.3) |
| PCBM-AT | 98.2(0.7) | 98.3(0.4) | 97.4(2.1) | 98.4(1.3) | 98.4(0.1) | 98.4(0.1) | 97.5(0.2) | 98.5(0.3) | 97.4(0.8) |
| ProbCBM-AT | 98.1(0.2) | 98.4(0.2) | 98.3(0.7) | 98.2(0.1) | 98.2(0.7) | 98.4(0.8) | 98.2(0.5) | 98.1(0.5) | 98.2(0.6) |
| Label-free CBM-AT | 98.2(0.5) | 98.1(0.3) | 98.4(0.2) | 98.4(0.6) | 97.5(1.1) | 97.5(0.2) | 98.5(0.2) | 98.8(0.6) | 98.3(0.5) |
| ProtoCBM-AT | 98.2(0.2) | 97.9(0.8) | 98.2(0.4) | 98.6(0.7) | 98.4(0.7) | 98.4(0.4) | 98.3(0.2) | 98.4(0.1) | 98.6(0.7) |
| ECBMs-AT | 99.1(0.2) | 98.5(0.4) | 98.6(0.5) | 98.3(0.7) | 97.2(0.9) | 98.3(0.7) | 98.2(0.7) | 98.5(0.8) | 98.3(1.6) |
| LEN | 98.2(0.4) | 98.4(0.3) | 98.2(0.1) | 98.4(0.5) | 98.3(0.2) | 98.4(0.3) | 98.2(0.1) | 98.4(0.5) | 98.3(0.2) |
| DKAA | 98.4(0.1) | 98.6(0.2) | 97.9(0.1) | 98.2(0.2) | 98.3(0.1) | 98.6(0.2) | 97.9(0.1) | 98.2(0.2) | 98.3(0.1) |
| MORDAA | 98.6(0.1) | 98.3(0.2) | 97.9(0.2) | 98.5(0.7) | 98.1(0.3) | 98.3(0.2) | 97.9(0.2) | 98.5(0.7) | 98.1(0.3) |
| DeepProblog | 97.2(0.1) | 97.2(0.4) | 97.1(0.3) | 97.5(0.1) | 97.1(0.4) | 97.2(0.4) | 97.1(0.3) | 97.5(0.1) | 97.1(0.4) |
| MBM | 98.2(0.1) | 98.2(0.3) | 98.1(0.2) | 98.5(0.1) | 98.1(0.4) | 98.2(0.3) | 98.1(0.2) | 98.5(0.1) | 98.1(0.4) |
| C-HMCNN | 98.2(0.1) | 98.2(0.2) | 98.1(0.3) | 98.5(0.6) | 98.1(0.2) | 98.2(0.2) | 98.1(0.3) | 98.5(0.6) | 98.1(0.2) |
| AGAIN | 98.3(0.5) | 98.2(0.6) | 98.1(0.7) | 98.2(0.1) | 98.2(0.6) | 98.5(0.3) | 98.7(0.4) | 98.2(0.7) | 98.3(1.1) |

**Erasure Attacks.** Erasure attacks attempt to subtly remove a specific concept without changing category prediction results. Gaps in perception and missing concepts are confusing for an analyst and very difficult to detect. For CBMs, we typically have a pre-set threshold $\gamma$ for determining whether a concept is an activated concept. Specifically, for a sample $x$, We use $h_c^{(m)}(x)$ to denote the activation value of the $m$-th concept output by the concept predictor $h_c(\cdot)$. If $h_c^{(m)}(x) - \gamma > 0$, then the $m$-th concept is an activated concept. Attackers learn adversarial perturbations for executing erasure attacks by the following objective function:

$$\text{MAX} \sum_{m \in M_E} \left( (\mathbb{I}[\gamma - h_c^{(m)}(x + \delta)] - \mathbb{I}[\gamma - h_c^{(m)}(x)]) \right) ,$$
$$\text{s.t.} \quad \text{argmax} \ h_y(h_c(x + \delta)) = \text{argmax} \ h_y(h_c(x)) \tag{23}$$

where $\mathbb{I}(\cdot)$ denotes the indicator function, $h_c(\cdot)$ is the concept predictor, and $h_y(\cdot)$ is the category predictor (task predictor). $\delta$ is the learnable perturbation. $M_E$ denotes the concept set that the attacker wishes to delete away.

**Introduction Attacks.** The purpose of introduction attacks is to manipulate the presence of irrelevant concepts without modifying the classification results. Such attacks hinder accurate analysis of model explanations. Attackers attempt to introduce new irrelevant concepts that do not previously exist in the concept set of the original sample. Unlike erasure attacks, introduction attacks require to raise the activation value of irrelevant concepts above the threshold $\gamma$. Therefore, the objective function of introduction attacks is represented as follows:

$$\text{MAX} \sum_{m \in M_I} \left( (\mathbb{I}[h_c^{(m)}(x + \delta) - \gamma] - \mathbb{I}[h_c^{(m)}(x) - \gamma]) \right)$$
$$\text{s.t.} \quad \text{argmax} \ h_y(h_c(x + \delta)) = \text{argmax} \ h_y(h_c(x)), \tag{24}$$

where $M_I$ denotes the set of concepts that the attacker wishes to introduce.

**Confounding Attacks.** Confounding attacks build on introduction attacks and erasure attacks. The confounding attack simultaneously removes relevant concepts and introduces irrelevant concepts. The confounding attack is a more powerful attack than the erasure attack and the introduction attack as it allows arbitrary tampering with the concept set of the original sample. The objective function of confounding attacks is as follows:

$$\text{MAX} \sum_{m \in M_E} \left( (\mathbb{I}[\gamma - h_c^{(m)}(x + \delta)] - \mathbb{I}[\gamma - h_c^{(m)}(x)]) \right)$$
$$+ \sum_{m \in M_I} \left( (\mathbb{I}[h_c^{(m)}(x + \delta) - \gamma] - \mathbb{I}[h_c^{(m)}(x) - \gamma]) \right)$$
$$\text{s.t.} \quad \text{argmax} \ h_y(h_c(x + \delta)) = \text{argmax} \ h_y(h_c(x)). \tag{25}$$

In this paper, confounding attacks are utilized to disrupt AGAIN, which contributes to a more complete evaluation on the comprehensibility of the explanations generated by AGAIN when concepts are missing and confused.

