# OpenReview forum: "Factor Graph-based Interpretable Neural Networks"
_ICLR.cc/2025/Conference — ICLR 2025 Poster_

### Official Review · Reviewer_YnPb · 2024-10-29

**Soundness:** 3
**Presentation:** 3
**Contribution:** 2
**Rating:** 6
**Confidence:** 4

**Summary:**

In this paper, the authors propose a novel method called AGAIN to generate comprehensible explanations under unknown perturbations by employing a factor graph approach. The method presents a significant contribution to the field, addressing an important gap with a new perspective. The paper is well-structured, and the authors have provided thorough theoretical justifications and experimental analyses. These analyses effectively demonstrate the superiority of the proposed method over existing ones.

**Strengths:**

The paper offers a well-motivated and clear presentation of a unique approach. The use of a factor graph to handle perturbations is innovative and highly relevant to the research community. Furthermore, the authors have provided sufficient theoretical support and empirical evidence to justify the effectiveness of their approach.

**Weaknesses:**

1. Clarification on Figure 3:
   In Figure 3, does the final output include hat{c}_{re} and prediction? Does the hat{c}_{re} mean the interpretation? Additionally, the figure lacks clarity on how hc is generated, which is crucial for understanding the method. Adding details on hc generation would make the figure more comprehensive.

2. Comparison Metrics:
   The authors compare their method with all baselines using the LSM metric. However, they only compare their method with CBM in terms of accuracy (P-ACC and E-ACC). It would be beneficial to extend the accuracy comparison to include all baselines to provide a more complete evaluation of the method’s performance.

3. Figure 7 Interpretation and Readability:
   In Figure 7, for each example, the authors provide two bar charts. Does the left bar chart represent the initial interpretation, and the right bar chart represent the interpretation combined with the factor graph? Clarification on this aspect would enhance the understanding of the figure. Additionally, some symbols and text in Figure 7 are overlapping.

4. Inconsistency in Line 463 and Table 5:
   The authors mention in line 463 that their method is compared with two baselines on the Synthetic-MNIST dataset. However, Table 5 lists four baselines. Furthermore, "ProbCBM" in Table 5 should be corrected to "ProCBM" for consistency. It is recommended that the authors proofread the paper to eliminate such inconsistencies and typographical errors.

**Questions:**

See Weaknesses

---

> ### Author Response · Authors · 2024-11-20
> **Response to Reviewer YnPb**
>
> We appreciate your valuable feedback, which are insightful and can improve the quality of our work to a great extent. Definitely, you point out some issues in our paper, giving us inspiration on how to revise the manuscript. In response to your comments, we have provided detailed answers and revised the manuscript accordingly.
>
> **Weaknesses 1:**
>
> We have made additions in the Rebuttal Revision of the paper **(Figure 3)**. The final output include $\hat{c}_{re}$ and prediction. $\hat{c}_re$ is the corrected explanation.
>
> **Weaknesses 2:**
>
> Following your comments, we revised the manuscript to provide more experimental results about accuracy comparison. We have made additions in the Rebuttal Revision of the paper **(Line 1242-1330 and Line 438-465)**.
>
> **Weaknesses 3:**
>
> We appreciate your constructive suggestions. Yes, the left bar chart represent the initial interpretation, and the right bar chart represent the interpretation combined with the factor graph. We revised the Rebuttal Revision of the paper **(Line 506)**.
>
> **Weaknesses 4:**
>
> We apologize for these typos. We have made additions in the Rebuttal Revision of the paper **(Line 087 and Line 312)**.

---

> > ### Comment · Reviewer_YnPb · 2024-11-26
> >
> > Thanks for the response and update! These do not affect my original overall review and I keep the original rating.

---

> > > ### Author Response · Authors · 2024-11-26
> > > **Response to follow-up comment by Reviewer YnPb**
> > >
> > > Thank you very much for your time and effort in reviewing our paper. If there are any other issues remaining, we are pleased to address them further.

---

### Official Review · Reviewer_SU9s · 2024-11-03

**Soundness:** 3
**Presentation:** 3
**Contribution:** 3
**Rating:** 6
**Confidence:** 3

**Summary:**

This paper proposes AGAIN, a neural network model that generates comprehensible explanations under unknown perturbations by integrating logical rules directly during inference, rather than relying on adversarial training. Using a factor graph to identify and rectify logical inconsistencies, AGAIN demonstrates superior interpretability and robustness compared to existing methods.

**Strengths:**

1. The paper presents an innovative method, AGAIN, that combines factor graphs with concept-level explanations to improve model interpretability under unknown perturbations.
2. The authors evaluate AGAIN across multiple datasets and baseline models, providing a broad view of its effectiveness.
3. The paper is clear and well-structured.

**Weaknesses:**

1. The factor graph requires predefined logical rules, which could be challenging to construct or generalize across different domains.
2. The algorithm’s process of exploring all possible intervention options to find the optimal solution could create computational overhead.

**Questions:**

1. The authors employed black-box perturbations to test robustness. Adding input noise could provide more convincing evidence.
2. I am concerned about the potential impact of large factor graphs on computational efficiency. I would like to see an analysis on this problem.
3. Currently, the algorithm explores all intervention options to select the optimal one. Is it possible to employ a simplified strategy, such as a heuristic or greedy algorithm, to reduce computational load?

---

> ### Author Response · Authors · 2024-11-20
> **Response to Reviewer SU9s**
>
> **Weaknesses 1:**
>
> This is indeed a limitation and thank you for your valuable comments. In fact, not only factor graphs require predefined logic rules, but all methods based on external knowledge integration inevitably require predefined knowledge. Therefore, how to automatically extract knowledge from the external environment is a separate and important problem. We have already discussed this limitation in the conclusion. **(Line 537-539)**
>
> **Weaknesses 2:**
>
> In theory, the computational overhead of AGAIN's intervention is indeed high. However, in practice, the number of interventions is usually no more than 100, because erroneous concepts are typically few in number. For example, in the CUB dataset, the number of incorrect concepts is usually only 1-10 [1]. Intervention operations are performed on a small number of concepts, as these erroneous concepts are localized during the detection phase. Even so, reducing the computational overhead of interventions is a valuable topic for discussion. Additionally, we have added a set of comparison experiments **(Line 1173-1208)** to address Question 3.
>
> **Questions 1:**
>
> Thank you for your comment. Indeed, the process of injecting black-box perturbations into input data is by nature adding input noise.
>
> **Questions 2:**
>
> Following your comments, we revised the Rebuttal Revision to provide more details about computational efficiency analysis **(Line 1143-1171)**.
>
> **Questions 3:**
>
> These strategies can reduce overhead in theory, but their practical payoff is not as significant. For further details, please refer to the experimental analysis in the Rebuttal Revision **(Line 1173-1208)**.
>
> [1] Understanding and enhancing robustness of concept-based models. AAAI.

---

> > ### Comment · Reviewer_SU9s · 2024-11-26
> >
> > Thank you for your response and the update! I will maintain the original rating.

---

> > > ### Author Response · Authors · 2024-11-26
> > > **Response to follow-up comment by Reviewer SU9s**
> > >
> > > Thank you very much for your time and effort in reviewing our paper. Your insightful suggestions greatly help us improve the quality of our paper. It is truly valuable for us to have such a meaningful discussion with you. If there are any other issues remaining, we are pleased to address them further.

---

### Official Review · Reviewer_PFRk · 2024-11-04

**Soundness:** 3
**Presentation:** 2
**Contribution:** 3
**Rating:** 5
**Confidence:** 4

**Summary:**

This paper proposes AGAIN (fActor GrAph-based Interpretable Neural Network), a new approach to maintain comprehensible neural network explanations when inputs are affected by unknown perturbations. AGAIN builds factor graphs from explanations, and integrates logical rules through the graphs. The system detects adversaries by identifying violations of real-world logic, and uses an interactive intervention strategy to fix incorrect explanations. Tested on three datasets (CUB, MIMIC-III EWS, and Synthetic-MNIST), AGAIN outperformed existing methods in generating comprehensible explanations under both known and unknown perturbations.

**Strengths:**

1. The paper proposes a novel and interesting idea that uses the logical correctness of explanation to detect and defense against noises and adversaries.
2. The three stage framework is reasonable to me.
3. The examples used in the paper are intuitive.

**Weaknesses:**

1. The concept bottleneck model is not easy to scale, as it requires manual annotations of concepts.
2. The implementation details of Section 4.2 is not very clear (the introduction is too conceptual). For example, is $\mathcal{G}$ a graph or a model? At the beginning I thought $\mathcal{G}$ is a graph, but in Line 221 it "reasons about a conditional probability".
3. The datasets used in this work are not very strong. I doubt if the work is applicable to real-world situations. At least, the datasets used cannot reflect adversarial scenarios in practice.

**Questions:**

How is the reasoning conducted on G? How is the probability estimation implemented in Equation 1? Please provide more details.

---

> ### Author Response · Authors · 2024-11-20
> **Response to Reviewer PFRk (Part 1)**
>
> **Weaknesses 1:**
>
> We fully understand your concern about manual annotations. However, this is widely considered as a limitation of datasets, rather than of CBMs [2]. CBM itself does not require an annotation process; therefore, AGAIN simply utilizes the well-established concept labels available in datasets. For concept annotation of datasets, many studies have already designed plug-ins to automatically annotate concepts specifically for datasets without concept labels [1-4]. These plug-ins use a multimodal-based approach to automatically extract concepts from samples and have achieved excellent performance. These plugins are sufficient to provide data support for CBMs training.
>
> **Weaknesses 2:**
>
> We apologize for any confusion Section 4.2 may cause. We have made an adjustment to Section 4.2 in the Rebuttal Revision **(Line 230-246)**. The phrase 'reasons about a conditional probability' is a typographical error, and $G$ refers to a graph.
>
> **Weaknesses 3:**
>
> (1) The datasets selected in this paper reflect adversarial scenarios in practice. For example, CUB has been a standard dataset for evaluating the prediction accuracy and interpretability of CBMs under adversarial attacks [5-8].
>
> (2) AGAIN is applicable to real-world situations. AGAIN was evaluated on real datasets capable of modeling real-world situations. Please allow us to give some existing examples to prove that these datasets are capable of modeling real situations, verifying the applicable of our model:
>
> * Mazumder et al. [10] argue that CUB can effectively evaluate “the performance of deep models in real life situations”.
>
> * Jiang et al. [11] considered the CUB dataset as a “fine-grained dataset in real-world applications”, and argued that evaluating on CUB "can contribute to advancement of the proposed approach in real-world scenarios".
>
> * As presented in [9], "the within-class variance in the CUB dataset is large due to variations in lighting, background, and extreme pose variations (e.g., flying birds, swimming birds, and perched birds partially obscured by branches)". These characteristics provide a strong simulation of real-world situations.
>
> (3) Real-world adversarial scenarios can be summarized into two types: 1) sensor disruption: an attacker can remotely disturb data sensors to perturb the data signals fed to the model (e.g., disrupting a patient's e-health metrics); 2) data tampering: an attacker can directly modify the data by adding a perturbation and re-feeding it back to the model (e.g., putting a disruptive texture on a road signage to misdirect the in-vehicle camera). The real datasets we use can fully cover the above two realistic adversarial scenarios. Specifically, the samples of the MIMIC -III EWS dataset which are injected with perturbations can simulate the sensor signals that are interfered by the attacker in Scenario 1. The samples of the CUB dataset injected with perturbation can simulate the disruptive texture imposed on the image by the attacker in Scenario 2.
>
> **Questions:**
>
> Following your comments, we revised the manuscript to provide more details about probability estimation. Please refer to **Line 230-246** in the Rebuttal Revision of the paper.
>
> Firstly, after each variable (concept and category) in $\mathcal{G}$ is assigned a value, boolean values are output from potential functions of all factors. These boolean values indicate whether the assignments of concepts and categories satisfy the logical rules represented by potential functions. Therefore, the weighted sum of all potential functions quantifies the extent to which concept assignments satisfy the logic rules in $\mathcal{G}$.
> Then, we seek to obtain the likelihood of the current concept assignments occurring, conditional on the known categories and logic rules. We quantify this likelihood by computing a conditional probability using the weighted sum of potential functions. We consider all possible concept assignments and compute the expectation of current concept assignments. This expected value is considered as the conditional probability, which is then used to detect whether concept activations are perturbed.
> For illustrative purposes, we provide an example. Suppose there are concepts $A$ and $B$. The current concept assignment is {1,0} denoting $A=1$ (active) and $B=0$ (inactive). We iterate through all four possible assignments: {1,0}, {0,1}, {1,1}, {0,0}. We compute the weighted sum of the potential functions for each of the four cases and compute the expectation of the potential function for {1,0}. This expectation is the conditional probability that concept assignment $\{1,0\}$ occurs conditionally on the known categories and logic rules.

---

> > ### Comment · Reviewer_PFRk · 2024-11-25
> >
> > 1. I am more curious about the "automatically annotated concepts" you mentioned in your rebuttal. This would be more fancinating than using established concepts.
> > 2. The current version of Sec 3, Sec 4.1, 4.2 are still not good. My suggestions are as follows. (1) Make the current writing more concise. Try avoid relying too much on ChatGPT in your writing, especially those that involve math or notations. (2) Provide a more detailed version of preliminary about PGM, as this is the fundamental model in your work. (3) Make it more clear how PGM provide explainable prediction based on the preliminary.
> >
> > I can reduce my confidence to 3. But I am not going to increase my score. I want to see other reviewers' comments.

---

> > > ### Author Response · Authors · 2024-11-25
> > > **Response to follow-up comment by Reviewer PFRk**
> > >
> > > Thank you very much for your response to our rebuttal. We are pleased to provide further clarification.
> > >
> > > * The mainstream paradigm for automatic concept annotation is to leverage large-language models (LLMs) such as GPT-3 to generate class-specific concepts and vision-language models (VLMs) such as CLIP to learn the mapping from inputs to concepts in an attribute-label-free manner. In fact, most datasets can be applied with this technique [1]. Thus, AGAIN enables training and inference on most datasets without original concept labels.
> > >
> > > * Thank you for your insightful comment. We revised and refined Sec 3, Sec 4.1, 4.2. We provided a more detailed version of the preliminary about CBM (We think by “PGM” you mean “CBM”), and explained how CBM generates concept-level explanations. Specifically, the CBM is trained on data points $(x, C, y)$, where the input $x$ is labeled with both concepts $C$ and the target $y$. $C=${$c_0$,...,$c_i$} is a set of binary values, where the $i$-th element in the set denotes whether or not $x$ contains the $i$-th human-specified concept, with $c=1$ denoting containment and $c=0$ denoting non-containment. The CBM consists of two components, the concept predictor and the category predictor. The concept predictor receives the input $x$ and predicts concepts $\hat{C}$. The category predictor receives predicted concepts $\hat{C}$ and predicts the target $y$. The CBM takes the concepts $\hat{c}$ predicted in the inference as the explanation of the model. $\hat{c}$ answers which human-specified concepts are contained in input $x$. These concepts directly determine the final prediction $\hat{y}$.
> > > We will incorporate these descriptions into the camera-ready version.
> > >
> > > Your suggestion is valuable to us. We're pleased to have a further discussion with you if you still have any concerns or questions.
> > >
> > > [1] Discover-then-name: Task-agnostic concept bottlenecks via automated concept discovery. ECCV.

---

> > > > ### Comment · Reviewer_PFRk · 2024-11-27
> > > >
> > > > This reminds me another question. How are the "logical rules" obtained in this work?

---

> > > > > ### Author Response · Authors · 2024-11-27
> > > > > **Response to follow-up comment by Reviewer PFRk**
> > > > >
> > > > > Thank you for your comments.
> > > > >
> > > > > Our logical rules are predefined. Specifically, for the CUB dataset, we used the table containing the correlations between concepts and categories, which was provided with the original dataset and annotated by ornithologists, to extract logical rules [1].
> > > > > For the MIMIC-III EWS dataset, we directly utilized the medical concepts and logical rules developed by [2]. These medical concepts and logical rules are obtained by medical experts who analyze the vital signs of samples.
> > > > > For the synthetic dataset, following [3], logical rules for handwritten digit concepts and categories are specified and constructed.
> > > > >
> > > > > In fact, in the original version of the paper, we had already included the rule extraction process for each dataset in the **Appendix C.1** **(Line 964-1021)**.
> > > > >
> > > > > Thank you very much for your time and effort in reviewing our paper. Hope we have well addressed your concerns. If there are any other issues remaining, we are pleased to address them further.
> > > > >
> > > > > [1] Concept bottleneck models. ICML2020.
> > > > >
> > > > > [2] Addressing leakage in concept bottleneck models. NeurIPS2022.
> > > > >
> > > > > [3] Probabilistic concept bottleneck models. ICML2023.

---

> > > > > ### Author Response · Authors · 2024-11-29
> > > > > **Official Comment by Reviewer PFRk**
> > > > >
> > > > > Dear reviewer PFRk,
> > > > >
> > > > > We really appreciate your effort in reviewing our paper, your suggestions are highly valuable to us. As the rebuttal phase is nearing its end, we would like to kindly follow up to ensure there is sufficient time for any further clarifications or concerns you might have. Please let us know if there is anything further we can do to assist or elaborate on.
> > > > >
> > > > > Thank you once again for your time and effort in reviewing our paper.
> > > > >
> > > > > Best regards,
> > > > >
> > > > > The authors of submission 7228

---

> ### Author Response · Authors · 2024-11-20
> **Response to Reviewer PFRk (Part 2)**
>
> **References:**
>
> [1]Label-free concept bottleneck models. ICLR.
>
> [2]Discover-then-name: Task-agnostic concept bottlenecks via automated concept discovery. ECCV.
>
> [3]Incremental Residual Concept Bottleneck Models. CVPR.
>
> [4]Semi-supervised Concept Bottleneck Models. ICML.
>
> [5]Understanding and enhancing robustness of concept-based models. AAAI.
>
> [6]Concept bottleneck models. ICML.
>
> [7]Addressing leakage in concept bottleneck models. NeurIPS.
>
> [8]Interactive concept bottleneck models. AAAI.
>
> [9]The caltech-ucsd birds-200-2011 dataset.
>
> [10] Few-shot lifelong learning. AAAI2021.
>
> [11] Navigating Real-World Partial Label Learning: Unveiling Fine-Grained Images with Attributes. AAAI2024.

---

> ### Comment · Reviewer_PFRk · 2024-12-02
>
> After going through the paper details with rebuttal information, I think this paper clearly does not reach the ICLR acceptance bar. Relying on pre-defined concepts and logics is the major concern. The approach is not novel or practically scalable. Another minor problem is that the paper's presentation still needs to go through some major revision before being considered publishable.

---

> > ### Author Response · Authors · 2024-12-03
> > **Response to follow-up comment by Reviewer PFRk**
> >
> > Thank you for your time and effort in reviewing our paper. In response to your comments, we provide further clarification:
> >
> > * We would like to emphasize that **how to automatically extract rules and concepts is not the focus of this paper but rather another problem worth investigating**. Most explanatory methods based on external knowledge are implemented by utilizing the predefined knowledge from domain experts [1]. Therefore, our paper follows this way to inject external knowledge. Notably, the challenge solved in this paper is how to improve the comprehensibility of explanations under unknown perturbations. To solve this new challenge, we directly utilize logical rules (predefined or extracted) to detect and correct explanation errors. Therefore, **AGAIN is certainly novel in terms of improving the comprehensibility of explanations**. Further, there are many approaches to automatically extract external knowledge or logical rules from datasets [2,3]. Similarly, there are many approaches to automatically annotate concepts for datasets [4,5,6,7]. These methods can provide effective external knowledge support and training data support for AGAIN. Thus, **the significance of AGAIN would not be diminished because it utilizes predefined rules**. Furthermore, we already provided the analysis on the scalability and computational efficiency of AGAIN (Appendix D.2). The experimental results show that the scalability of AGAIN is good.
> >
> > * We appreciate your valuable feedback on presentation, which is insightful and can improve the quality of our work to a great extent. We will thoroughly review the final manuscript and revise misleading or unclear sentences and paragraphs to improve our presentation.
> >
> > Thanks again for your comments. Hope we have well addressed your concerns.
> >
> > [1] Harnessing Prior Knowledge for Explainable Machine Learning: An Overview. IEEE SaTML2023.
> >
> > [2] Simre: Simple contrastive learning with soft logical rule for knowledge graph embedding. Information Sciences 2024.
> >
> > [3] Collaborative artificial intelligence system for investigation of healthcare claims compliance. Scientific Reports 2024.
> >
> > [4] Label-free concept bottleneck models. ICLR.
> >
> > [5]Discover-then-name: Task-agnostic concept bottlenecks via automated concept discovery. ECCV.
> >
> > [6]Incremental Residual Concept Bottleneck Models. CVPR.
> >
> > [7]Semi-supervised Concept Bottleneck Models. ICML.

---

### Official Review · Reviewer_XLZi · 2024-11-04

**Soundness:** 3
**Presentation:** 3
**Contribution:** 3
**Rating:** 8
**Confidence:** 4

**Summary:**

The paper introduces AGAIN (fActor GrAph-based Interpretable Neural Network), which generates comprehensible explanations for neural network decisions even under unknown perturbations. Unlike traditional adversarial training methods, AGAIN incorporates logical rules through a factor graph to identify and rectify explanatory logical errors during inference. The model is validated on three datasets: CUB, MIMIC-III EWS, and Synthetic-MNIST, demonstrating superior performance compared to existing baselines

**Strengths:**

- **Good presentation and structure** The paper is well-structured and easy to read. The mathematical definition of the proposed method is clear and well-defined.
- **Nice experimental campaign**: Extensive experiments on three datasets (CUB, MIMIC-III EWS, and Synthetic-MNIST) demonstrate the superior performance of AGAIN over the compared baselines, although these results hold mainly for a metric (LSM) that has been created ad-hoc.

**Weaknesses:**

## Major Issues

- **Novelty, Related work and Compared methods**: the main issue with the current paper is that it only considers methods injecting knowledge into the models by means of factor graphs. However, the integration of knowledge into model predictions has been fully studied under different point of views: probabilistic (e.g., DeepProblog [1] and all its variants), logic constraints (see the survey of Giunchiglia et al. [2]). Also, it is not the first method defending against adversarial attack with knowledge and without employing adversarial training. Some of these methods have been already employed to defend against adversarial attacks, such as [3-4]. [5] is a survey entirely dedicated to enhancing interpretability and adversarial robustness with prior knowledge. [6] has shown in the context of concept-based models that it can learn logic rules at training time and use them at inference time to defend against adversarial attacks. This is also reflected in the experimental campaign that is extensive but does not consider any methods injecting prior knowledge to defend against adversarial attacks. The only compared methods are CBMs or adversarial-trained versions of the same models.


- **Paper positioning and Preliminaries**: the method provides explanations and a defence mechanism that is based on concept predictions; thus, it applies only to concept-based models. Most of the compared methods also belongs to this niche. While this is fully acceptable, explicit references to concept-based models only appears in Section 3-4. Therefore, I think it should state earlier that this is the focus of the paper, as most of the related work mentioned in the paper does not focus on concept-based explanations. Furthermore, there is no explicit mentions to concept-based models in the preliminaries. The “Interpretable Neural Network” paragraph should include citations to this literature and explain concept-based models.

## Minor Issues
- P.2 “[…] even if the adversarial samples are available, retraining is only effective for at most one perturbation type” I think this statement is quite strong, and although it may have been proved for some attacks in Tan & Tian, 2023, I don’t think it is a general assumption that we can make. I think this sentence should be soften to “retraining is effective only for few perturbation types at the same time”.
- P.2 “[…] to ensure the expectation of the potential function is in its maximum value”. It is not clear at this point what is the potential function. Consider rephrasing this sentence.
- P.2 “The explanations that are further regenerated align with the exogenous logic.” Not clear in this case what are the exogenous factors.
- P.2-3: The term "defenses against comprehensibility" seems a bit misleading. It implies that the goal is to prevent explanations from being understandable, which is not the case. Instead, the focus is on defending the comprehensibility of explanations against perturbations. “defenses of comprehensibility” colud be a more appropriate definition.


[1] Robin Manhaeve, Sebastijan Dumancic, Angelika Kimmig, Thomas Demeester, Luc De Raedt: DeepProbLog: Neural Probabilistic Logic Programming. NeurIPS 2018: 3753-3763

[2] Giunchiglia, E., Stoian, M. C., & Lukasiewicz, T. (7 2022). Deep Learning with Logical Constraints. In L. D. Raedt (Ed.), Proceedings of the Thirty-First International Joint Conference on Artificial Intelligence, IJCAI-22 (pp. 5478–5485). doi:10.24963/ijcai.2022/767

[3] Yin, M., Li, S., Cai, Z., Song, C., Asif, M. S., Roy-Chowdhury, A. K., & Krishnamurthy, S. V. (2021). Exploiting multi-object relationships for detecting adversarial attacks in complex scenes. In proceedings of the IEEE/CVF international conference on computer vision (pp. 7858-7867).

[4] Melacci, Stefano, et al. "Domain knowledge alleviates adversarial attacks in multi-label classifiers." IEEE Transactions on Pattern Analysis and Machine Intelligence 44.12 (2021): 9944-9959.

[5] Mumuni, Fuseini, and Alhassan Mumuni. "Improving deep learning with prior knowledge and cognitive models: A survey on enhancing interpretability, adversarial robustness and zero-shot learning." Cognitive Systems Research (2023): 101188.

[6] Ciravegna, Gabriele, et al. "Logic explained networks." Artificial Intelligence 314 (2023): 103822.

**Questions:**

- Can the authors provide a comparison against one of the methods injecting knowledge (prior or learnt)?
- Scalability: The two limitations that have been reported (domain knowledge changes, correct prediction categories) are non-negligible. How could this method be extended to face these limitations?

---

> ### Author Response · Authors · 2024-11-20
> **Response to Reviewer XLZi**
>
> **Weaknesses Major Issues 1:**
>
> Thank you for your comment. Here, we’d like to highlight:
>
> (1) In this paper, we focus on generating comprehensible explanations under unknown perturbations instead of injecting knowledge into the model. To address this problem, we require an effective way to inject external knowledge. We demonstrate in a supplemental ablation analysis **(Line 366-375)** that knowledge injected through factor graphs is more helpful in generating comprehensible explanations compared to other knowledge integration methods. After injecting external knowledge, we design an error detection mechanism and a concept intervention strategy to identify and correct wrong concept activations, thereby improving the comprehensibility of explanations.
>
> (2)Although AGAIN also injects external knowledge, it has a distinctly different research goal and research question compared to [1-6]. Specifically, [1-4] focus on constraining the model to output correct predictions, rather than ensuring the model produces correct explanations under adversarial attacks. As a result, the model may still produce incorrect explanations even when it predicts correctly. [5] highlights that external knowledge can enhance interpretability, but it does not explain how the model ensures interpretability when explanations are disturbed by unknown perturbations. Similarly, LEN [6] learns rules between concepts and categories to correct predicted categories, but it does not address the correction of concepts. Therefore, we believe that none of these studies tackle the issue discussed in our paper.
>
> (3)You raised concerns about the sufficiency of the comparison methods, which suggests that other readers may be similarly confused. Therefore, we have added experiments to compare the performance differences between AGAIN and the methods listed by you, focusing on knowledge integration. This experiment was included in the Rebuttal Revision of the paper **(Line 366-375)**. The results show that the knowledge injected by the existing methods relies solely on forward reasoning to correct predictions and cannot reverse the conditional probabilities of the explanations based on those predictions. As a result, these methods lack the ability to correct explanations. In this regard, factor graphs offer an irreplaceable advantage.
>
> **Weaknesses Major Issues 2:**
>
> We appreciate your constructive suggestions. Based on your suggestions, we have added concept-based models to the related work and preliminaries sections. These additions can be found in the Rebuttal Revision of the paper **(Line 117-121 and Line 140-145)**.
>
> **Weaknesses Minor Issues:**
>
> We have made additions in the Rebuttal Revision of the paper **(Line 056, Line 082, Line 083, and Line 103)**.
>
> **Question 1:**
>
> We added experiments to compare the performance differences between AGAIN and knowledge-based integration methods. These experiment were included in the Rebuttal Revision of the paper **(Line 366-375)**.
>
> **Question 2:**
>
> We plan to extend AGAIN to overcome the above limitations from two aspects:
>
> (1) Domain knowledge changes: we will design modules to automatically mine logical rules, such as automatically constructing knowledge graphs, mining logical rules from knowledge graphs, and adaptively learning external knowledge from different scenarios.
>
> (2) Correct prediction categories: we will enhance the logical association between concepts and concepts to reduce the sensitivity of external knowledge to prediction categories.
>
> [1] DeepProbLog: Neural Probabilistic Logic Programming. NeurIPS.
>
> [2] Deep Learning with Logical Constraints. IJCAI.
>
> [3] Exploiting multi-object relationships for detecting adversarial attacks in complex scenes.CVPR.
>
> [4]Domain knowledge alleviates adversarial attacks in multi-label classifiers.TPAMI.
>
> [5] Improving deep learning with prior knowledge and cognitive models: A survey on enhancing interpretability, adversarial robustness and zero-shot learning.Cognitive Systems Research.
>
> [6] Ciravegna, Gabriele, et al. "Logic explained networks. AI.

---

> > ### Comment · Reviewer_XLZi · 2024-11-21
> >
> > I thank the authors for taking the time for answering my questions and producing further experiments.
> >
> > However, I still have some questions:
> > 1) Why do you make comparisons in terms of LSM only? This is not a standard metric and it is introduced only in the appendix. I think this is a **major issue** in the current paper that I did not notice during the first reading. Can you provide more explanations regarding this metric, in particular for how you compute it for other methods? You say that LSM consider the weighted sum of potential functions within the factor graph, but how can you compute it for the compared methods that do not provide a factor graph?
> > 2) Can you provide more explanations about the other two metrics P-ACC and E-ACC? Can you provide some correlations w.r.t. standard CBM metrics like task accuracy and concept accuracy? Why are the results for the other methods for these metrics only reported in the appendix?
> > 3) Why are the results for the P-ACC similar for all methods? What does it mean that the perturbations do not affect the outcome? Normal adversarial attacks are indeed those that modify the outcome of a model
> > 4) Particularly if you are considering specific attacks that are not the standard one, can you provide more examples of the perturbations you consider, in terms of the adversarial attack employed to discover them?
> > 5) While I thank you for taking the time to test some of the suggested methods, since you have put these results only in the Appendix, I still do not find your comparison fair enough. The methods that you compare with have not been studied to be consistent against adversarial perturbations, while the suggested ones are.

---

> > > ### Author Response · Authors · 2024-11-22
> > > **Response to follow-up comment by Reviewer XLZi （Part 2）**
> > >
> > > **3.**
> > >
> > > (1)This is because the perturbations discussed in this paper only disturb the explanations (concepts) and not the results of task prediction, and therefore the P-ACC (task predictive accuracy) metric hardly changes under different perturbations. This also illustrates that the effectiveness of AGAIN in improving the comprehensibility of explanations is not reflected in whether or not the P-ACC metrics change, but primarily in changes in the E-ACC and LSM metrics. The reason we still report P-ACC is to demonstrate that AGAIN did not sacrifice task predictive accuracy in order to improve the quality of explanations.
> > >
> > > (2)We would like to emphasize that most research in the interpretability domain mainly focuses on defending against perturbations that affect the explanation without changing the prediction [6]. This focus is due to the fact that, when the predictions are incorrect, the pursuit of explanations becomes less meaningful [7]. This is because ambiguity arises only when correct predictions are paired with incorrect explanations. At the same time, we would also like to emphasize that such perturbations are already widespread and widely confirmed, and are not mentioned for the first time in this paper. For example, [5] claims that “An attacker can easily disrupt concepts without changing the prediction label.” A simple way to achieve this is to optimize an objective function to maximize the difference in concept activation while ensuring that the final prediction remains unchanged. This objective function converges with only a few optimization steps. The attacker can then search the solution space for perturbations that meet the condition. As a result, attacks based on these perturbations are easy to implement and widespread.
> > >
> > > **4.**
> > >
> > > Adversarial attacks that disrupt the explanation do differ from standard adversarial attacks in their goals. Here, we provide 3 perturbation examples to illustrate the specific implementation details of adversarial attacks:
> > >
> > > (1)Erasure perturbation
> > >
> > > The goal of the erasure perturbation is to subtly remove concepts from an explanation without changing the results of the task prediction. In practice, for CBM, we typically have a pre-set threshold a for determining whether a concept is an activated concept. Specifically, for a sample x, if $h_c^{(n)} (x)-a > 0$, then the nth concept is an activated concept. Let $N$ denote a set of concepts that the attacker wishes to remove. To remove the existence of these concepts, the attacker's goal is as follows:
> > >
> > > \begin{array}{l}
> > > Max{\rm{ }}\sum\limits_{n \in N} {(\mathbb{I}[a - {h_c}^{(n)}(x + \delta )] - \mathbb{I}[a - {h_c}^{(n)}(x)])} \\
> > > s.t.{\rm{ argmax }}{h_y}\left( {{h_c}\left( {x + \delta } \right)} \right) = {\rm{argmax }}{h_y}\left( {{h_c}\left( x \right)} \right)
> > > \end{array}
> > >
> > > where $\mathbb{I}()$ denotes the indicator function, $h_c()$ is the concept predictor, and $h_y()$ is the category predictor (task predictor). $\delta$ is the learnable perturbation.
> > >
> > > (2)Introduction perturbation
> > >
> > > The goal of Introduction perturbation is to allow the existence of irrelevant concepts without modifying the task prediction results. The goal of the attacker is as follows:
> > >
> > > \begin{array}{l}
> > > Max{\rm{ }}\sum\limits_{n \in N} {(\mathbb{I}[{h_c}^{(n)}(x + \delta ) - a] - \mathbb{I}[{h_c}^{(n)}(x) - a])} \\
> > > s.t.{\rm{ argmax }}{h_y}\left( {{h_c}\left( {x + \delta } \right)} \right) = {\rm{argmax }}{h_y}\left( {{h_c}\left( x \right)} \right)
> > > \end{array}
> > >
> > > (3)Confounding perturbation
> > >
> > > The goal of Confounding perturbation is to simultaneously remove relevant concepts and introduce irrelevant concepts. The goal of the attacker is the sum of the above two goals.
> > >
> > > Examples of the above three perturbations have been implemented by [10] with good attack results. This objective function converges with only a few optimization steps. The attacker can then search the solution space for perturbations that meet the condition.
> > >
> > > **5.**
> > >
> > > To further address your concerns about fairness, we have added these experimental results to the main paper **(Line 378-411)**. We have followed your comment that “Some of these methods have been already employed to defend against adversarial attacks, such as [3-4]”. “[6] has shown in the context of concept-based models that it can learn logic rules at training time and use them at inference time to defend against adversarial attacks.”. We have compared the methods suggested by the reviewer to specifically defend against adversarial perturbations: DKAA [8], MORDAA [9], and LEN [10]. The performance of AGAIN remains optimal compared to these three methods.

---

> > > ### Author Response · Authors · 2024-11-22
> > > **Response to follow-up comment by Reviewer XLZi （Part 3）**
> > >
> > > **References：**
> > >
> > > [1]Semi-supervised Concept Bottleneck Models. ICML2024.
> > >
> > > [2]Concept bottleneck models. ICML2020.
> > >
> > > [3]Addressing leakage in concept bottleneck models. NeurIPS2022.
> > >
> > > [4]Harnessing Prior Knowledge for Explainable Machine Learning: An Overview. SaTML2023.
> > >
> > > [5]Understanding and enhancing robustness of concept-based models. AAAI, 2023.
> > >
> > > [6]Adversarial attacks and defenses in explainable artificial intelligence: A survey. Information Fusion, 2024.
> > >
> > > [7]Interpretation of neural networks is fragile. AAAI, 2019.
> > >
> > > [8]Exploiting multi-object relationships for detecting adversarial attacks in complex scenes. CVPR2021.
> > >
> > > [9]Domain knowledge alleviates adversarial attacks in multi-label classifiers. IEEE Transactions on Pattern Analysis and Machine Intelligence. 2021.
> > >
> > > [10] Understanding and enhancing robustness of concept-based models. AAAI2023.

---

> > > ### Author Response · Authors · 2024-11-26
> > > **Official Comment by Authors**
> > >
> > > Dear reviewer XLZi,
> > >
> > > We really appreciate your suggestions and comments, which are highly valuable to us. We would like to kindly follow up to ensure there is sufficient time for any further clarifications or concerns you might have. Please let us know if there is anything further we can do to assist or elaborate on.
> > >
> > > Thank you once again for your time and effort in reviewing our paper and engaging in further discussions with us.
> > >
> > > Best regards,
> > >
> > > The authors of submission 7228

---

> > > ### Author Response · Authors · 2024-11-26
> > > **Response to follow-up comment by Reviewer XLZi （Part 4）**
> > >
> > > Dear reviewer  XLZi,
> > >
> > > Regarding Question 1 of your Official Comment, we’d like to provide more detailed clarification.
> > >
> > > We have already put the results of E-ACC and P-ACC for one real dataset into the main paper **(Line 438-466)**. The E-ACC and P-ACC for the other datasets are placed in the Appendix **(Line 1254-1330)**. The results indicate that AGAIN's performance remains superior in evaluations using standard metrics.
> > >
> > > We apologize for the insufficient clarification earlier.
> > >
> > > Thank you again for your time and effort in reviewing our paper and look forward to further discussions with you.

---

> > > > ### Comment · Reviewer_XLZi · 2024-11-26
> > > >
> > > > I would like to express my gratitude to the authors for their thorough work and the effort they put into addressing my concerns in this rebuttal. As a result, I have raised both my score and my confidence in the paper.
> > > >
> > > > While most of my concerns have been satisfactorily addressed, I still have one remaining issue regarding the adversarial attacks employed. The attacks reported in your response are fine, and I agree these adversarial attacks are not standard, as they aim to change the explanation rather than decrease the model's confidence. For this reason, however, I believe it is important that these attacks are presented in the revised version of the main paper. The name of the attack, along with a reference to the paper proposing it, should be included in the main text (the equation as well, if space permits; otherwise, it can be placed in the appendix).
> > > >
> > > > I apologize if I have overlooked these details. If the authors could add this information or point out where it has been included, I would be happy to further raise my score.

---

> > > > > ### Author Response · Authors · 2024-11-26
> > > > > **Response to follow-up comment by Reviewer XLZi**
> > > > >
> > > > > Thank you very much for your kind reply and raising your score and your confidence. We are glad to hear that our rebuttal has addressed your concerns and questions. Your thoughtful suggestions have greatly helped us improve the quality of our paper.
> > > > >
> > > > > * We have incorporated the names and descriptions of these adversarial attacks, along with references, into the Rebuttal Revision of the main paper **（Line 177-185）**. Additionally, we have placed a more detailed implementation containing descriptions of the formulas in the appendix  **（Line 1310-1354）**.
> > > > >
> > > > > * In fact, in the original version of the main paper, we also summarized existing adversarial attack methods against explanations in the Related Work Section **(Line 101-123)**.
> > > > >
> > > > > Thanks again for your comments. Hope we have well addressed your concerns. If there are any other issues remaining, we are pleased to address them further.

---

> > > > > ### Author Response · Authors · 2024-11-27
> > > > > **Response to follow-up comment by Reviewer XLZi**
> > > > >
> > > > > Dear reviewer XLZi,
> > > > >
> > > > > Thank you very much for your suggestion regarding the addition of introductions to adversarial attacks. We have revised the Rebuttal Revision following your suggestions. As the PDF revision phase is nearing its end, we would like to kindly follow up to ensure there is sufficient time for any further clarifications or concerns you might have. Please let us know if there is anything further we can do to assist or elaborate on.
> > > > >
> > > > > Thank you once again for your time and effort in reviewing our paper.
> > > > >
> > > > > Best regards,
> > > > >
> > > > > The authors of submission 7228

---

> > > > > ### Author Response · Authors · 2024-11-28
> > > > > **Response to follow-up comment by Reviewer XLZi**
> > > > >
> > > > > Thank you very much for raising your score from 6 to 8. Your insightful suggestions greatly help us improve the quality of our paper. It is truly valuable for us to have such a meaningful discussion with you.

---

> ### Author Response · Authors · 2024-11-22
> **Response to follow-up comment by Reviewer XLZi （Part 1）**
>
> Thank you for your prompt replies. The comments you have raised help us to improve the quality of our paper.
> Based on your comments, we have revised the Rebuttal Revision and provided detailed responses below.
>
> **1.**
>
> (1)We did not limit our comparison to LSM alone; we also evaluated standard CBM metrics, including E-ACC and P-ACC. These metrics are widely adopted in studies related to CBMs [1-3]. Due to space constraints, descriptions of all metrics, including the standard ones, were placed in the appendix, not just the LSM.
>
> (2)The LSM evaluates the degree to which concept-level explanations satisfy predefined logical rules. If a rule holds under the current concept prediction, the corresponding potential function for that rule has a value of 1, otherwise it is 0. We compute the weighted ratio of the potential functions with a value of 1 to all potential functions in the factor graph.The LSM is defined as the mean of the weighted ratios across all the test samples.
>
> (3)We compute the LSMs of other methods based solely on their generated concept activations. These activations are assigned to our factor graph, which is then used to compute the LSM. Notably, other methods do not need to construct a separate factor graph for this process. To ensure experimental fairness, all methods utilize the same factor graph during the LSM computation phase.
>
> **2.**
>
> Thank you for your further comment.
>
> (1)E-ACC and P-ACC denote the predictive accuracy of concepts and the predictive accuracy of tasks, respectively. Therefore, we can simply understand that E-ACC is the accuracy of explanation and P-ACC is the accuracy of final prediction. In datasets, each sample contains task prediction labels and concept prediction labels. Task prediction labels are stored as indexes of categories and concept prediction labels are stored as binary vectors. Each element of the vector indicates whether each concept is active or not (whether the concept exists in the sample). Therefore, we calculate the similarity of labels and predictions to compute the E-ACC and P-ACC. In fact, we have already described this in the paper **(Line 1054-1062)**, but we would like to clarify the reviewers' concerns by providing more details.
>
> (2)When there is no perturbation in the environment, concept predictive accuracy is positively correlated with task predictive accuracy. In CBMs, the input of the category predictor is concepts. Therefore, a high concept predictive accuracy enhances the higher the category predictive accuracy as well. However, in the adversarial perturbation setting, an attacker can tamper with several concepts that will not affect the final prediction, resulting in a significant reduction in concept accuracy while the prediction accuracy remains almost unchanged. Moreover, LSM is positively correlated with concept predictive accuracy because concept labels typically satisfy most critical logical rules.
>
> (3)In fact, we made a lot of efforts in order to put the comparison results of all metrics in the main paper. However, due to space constraints in the paper, we had to put the results of the comparison between E-ACC and P-ACC (this contains 6 tables) in the appendix. We will readjust the layout of the camera-ready version so that we can put the results of the E-ACC and P-ACC comparisons into the main paper. But even so, we would like to emphasize that the comprehensibility of explanations naturally depends on the extent to which they satisfy human perception ( the logical rules of the real world) [4]. Thus, the results of the LSM comparisons are a straightforward demonstration of the advantages of AGAIN in enhancing the comprehensibility of explanations.

---

### Official Review · Reviewer_XsjD · 2024-11-04

**Soundness:** 3
**Presentation:** 2
**Contribution:** 3
**Rating:** 6
**Confidence:** 3

**Summary:**

The paper proposed a method using factor graphs to correct errors in concept-based explanations caused by perturbations.
Specifically, it constructs a factor graph using predefined logical rules between concepts or between concepts and categories.
This graph helps identify logical errors in the output from concept-based interpretable neural networks by evaluating the likelihood of these errors.
Additionally, by leveraging the factor graph, it is possible to correct these logical errors in the explanations.
Experimental comparisons on three datasets demonstrate that the proposed method outperforms existing concept-based approaches.

**Strengths:**

- The proposed method enables both the detection of logical errors and the correction of these errors within a single framework.
- Compared to other concept-based interpretable neural networks, the proposed method achieves higher comprehensiveness in explanations, regardless of whether the perturbations are known or unknown.

**Weaknesses:**

- While the proposed method assumes that explanations change due to perturbations without affecting predictions, this seems unrealistic. Particularly in interpretable neural networks with a concept bottleneck structure, as assumed in this study, changes in the concepts outputted by the neural network would naturally lead to changes in predictions, which undermines this assumption.
- The proposed method requires predefined logic rules between concepts and categories. If these rules are already known, wouldn’t it be possible to detect inconsistencies between concepts and predictions without the need for the factor graph? The advantage of using a factor graph is unclear.
- As noted in the minor comments below, there is room for improvement in the writing.

Minor comments:
- The explanation of Figure 3 is insufficient.
- There is no reference to Figure 4.

**Questions:**

- Defining the logic rule set seems expensive. Would it be difficult to construct a factor graph by connecting concepts and categories with a bipartite complete graph and estimating the weights $w_i$?
- How do you distinguish between coexistence and exclusion in the factor graph?
- It would be better to describe the specific estimation algorithm of $w_i$ in the Appendix.
- Line 336-337: What is "attributional training"?

---

> ### Author Response · Authors · 2024-11-20
> **Response to Reviewer XsjD**
>
> **Weaknesses 1:**
>
> Thank you for your comments. We understand your concern about the reliability of our assumption. Actually, no. This assumption is realistic. Predictions are erroneous when they are affected by perturbations. We’d like to emphasize that most studies in the interpretability domain mainly focus on defending against perturbations that affect the explanation without changing the predictions [2]. This focus is due to the fact that, when the predictions are incorrect, the pursuit of explanations becomes less meaningful [3]. This is because ambiguity arises only when correct predictions are paired with incorrect explanations. Regarding your concern, we’d like to give a more detailed explanation of this situation:
> For concept bottleneck structures, attackers can generate perturbations that only interfere with concepts but not predictions. For example, [1] claims that “An attacker can easily disrupt concepts without changing the prediction label.” A simple way to achieve this is to optimize an objective function to maximize the difference in concept activation while ensuring that the final prediction remains unchanged. This objective function converges with only a few optimization steps. The attacker can then search the solution space for perturbations that meet the condition. As a result, attacks based on these perturbations are easy to implement and widespread.
>
> **Weaknesses 2:**
>
> External knowledge is typically expressed as logical rules. However, these rules are often discrete. Factor graphs can incorporate these discrete rules, enabling the detection and correction of wrong explanations. Known logic rule sets can only handle strictly deterministic reasoning (either established or unestablished). This limitation makes purely logic-based reasoning inaccurate, as real-world external knowledge is often uncertain or fuzzy. To accurately detect erroneous interpretations, it is essential to reason with uncertainty about this external knowledge. Factor graphs enable this by leveraging rule weight learning and conditional probability estimation, thereby enhancing detection accuracy.
> In contrast, while a collection of known logic rules can also detect erroneous explanations, its detection accuracy may be limited due to the absence of uncertainty reasoning. To validate this, we have included a set of ablation experiments in the Rebuttal Revision **(Line 1119-1133)**, comparing the effectiveness of factor graphs with purely logic rule sets. The experimental results demonstrate that factor graphs significantly outperform methods that rely solely on logic rules.
>
> **Weaknesses 3:**
>
> The explanation of Figure 3 is insufficient. There is no reference to Figure 4.
> Thank you for pointing out these issues. We have revised the manuscript based on your suggestions. You may refer to **Line 209** and **Figure 3**.
>
> **Questions 1:**
>
> Constructing a bipartite complete graph is a straightforward way. But the overhead of constructing a bipartite complete graph is the same as AGAIN. After constructing the bipartite complete graph, external knowledge still needs to be injected by setting weights based on logical rules. If logical rules are completely omitted and weights are learned solely from data samples, the graph fails to improve the comprehensibility of the explanation, as it lacks external knowledge necessary for detecting and correcting incorrect concepts. Therefore, to ensure the comprehensibility of the explanation under unknown perturbations, we must construct the factor graph using logical rules.
>
> **Questions 2:**
>
> Each factor is defined as a potential function that performs logical operations based on different rules, which can be categorized into coexistence and exclusion operations. We explained this point in the paper **(Line 216-221)** and apologize for any confusion caused by our initial explanation. Additional clarification has been included in the Rebuttal Revision **(Line 213-215)**.
>
> **Questions 3:**
>
> Thank you for your suggestion. We have made additions in the Rebuttal Revision of the paper **(Line 1097)**.
>
> **Questions 4:**
>
> Attributional training focuses on retraining the model with adversarial samples to maximize the similarity between the model's explanations and the explanatory labels. Therefore, attributional training can be considered a type of adversarial training. we have made adjustments in the Rebuttal Revision **(Line 358)**.
>
> [1] Understanding and enhancing robustness of concept-based models. AAAI, 2023.
>
> [2] Adversarial attacks and defenses in explainable artificial intelligence: A survey. Information Fusion, 2024.
>
> [3] Interpretation of neural networks is fragile. AAAI, 2019.
>
> [4] Enhanced regularizers for attributional robustness. AAAI, 2021.

---

> > ### Comment · Reviewer_XsjD · 2024-11-27
> >
> > Thank you for your response. My concerns have been addressed, and I would like to increase the scores accordingly.

---

> > > ### Author Response · Authors · 2024-11-28
> > > **Official Comment by Authors**
> > >
> > > Thank you very much for raising your rating. We are glad to hear that our rebuttal has addressed your concerns and questions. Your insightful suggestions greatly help us improve the quality of our paper. It is truly valuable for us to have such a meaningful discussion with you.

---

> ### Author Response · Authors · 2024-11-27
> **Official Comment by Authors**
>
> Dear reviewer XsjD,
>
> We really appreciate your effort in reviewing our paper, your suggestions are highly valuable to us. As the rebuttal phase is nearing its end, we want to kindly follow up to ensure there is sufficient time for any further clarifications or concerns you might have. Please let us know if there is anything further we can do to assist or elaborate on.
>
> Thank you once again for your time and effort in reviewing our paper.
>
> Best regards,
>
> The authors of submission 7228

---

### Author Response · Authors · 2024-12-02
**Summary of Revisions**

We sincerely appreciate the valuable feedback from all reviewers, which has significantly improved our work. In response, we have uploaded a revised version of the paper with substantial updates to address all the concerns and suggestions. Below, we provide a summary of the revisions made:

**1. Baseline addition:** Based on comments from reviewer XLZi, we added 6 baselines, including 4 typical methods based on knowledge integration and 2 methods based on perturbation defense.

**2. Computational efficiency analysis:** We added the experimental analysis of the computational efficiency of AGAIN. (Appendix D.2).

**3. Ablation analysis:** To enhance the evaluation on the validity of the factor graph, we added the ablation analysis of the factor graph. (Appendix D.1)

**4. Comparison of intervention strategies:** To evaluate the validity of intervention strategies, we added comparison experimental analysis and time complexity analysis of intervention strategies. (Appendix D.3)

**5. Comparisons of E-ACC and P-ACC:** We list the results of the E-ACC and P-ACC comparisons of AGAIN with all baselines on the three datasets. (Table 3, Table 4, Table 9, Table 10, Table 11, Table 12)

**6. Implementation details of adversarial attacks:** We provide implementation details of the adversarial attacks employed in our paper. (Appendix E)

**7. Weight estimation:** We provide implementation details of the learning process on factor weights. (Appendix C.5)

**8. Paragraph modification and section reorganisation:** We have thoroughly reviewed the manuscript and revised misleading or unclear sentences and paragraphs. We added the definition of adversarial attacks in Section 3.

**9. Figure adjustment:** We revised Figure 3 for clarity of the concept intervention process. We revised Figure 7 to eliminate the issue of overlapping elements in the figure.

We believe these updates address the reviewers' concerns and substantially improve the quality of the paper. We thank all reviewers for their constructive feedback and support.

---

### Meta-Review · Area_Chair_FdRQ · 2024-12-17

**Metareview:**

The paper proposes AGAIN (fActor GrAph-based Interpretable Neural Network), a method that uses factor graphs to correct logical errors in concept-based explanations caused by unknown perturbations. By integrating predefined logical rules between concepts and categories, the method identifies and rectifies inconsistencies in the outputs of concept-based neural networks. The approach demonstrates its effectiveness through experiments on three datasets—CUB, MIMIC-III EWS, and Synthetic-MNIST—outperforming existing baselines in generating comprehensible explanations under perturbations.

Strengths

+ The method enables both the detection and correction of logical errors within a single framework.
+ AGAIN achieves superior interpretability compared to other concept-based neural networks, even when perturbations are unknown.
+ The mathematical framework is well-defined, and the paper is clear and well-structured overall.
+ Extensive experiments on three datasets demonstrate the method's superior performance compared to existing baselines.
+ The examples provided are intuitive, and the three-stage framework is reasonable.

Weaknesses

+ The method assumes explanations change under perturbations without affecting predictions, which may not hold true for concept bottleneck models.
+ The reliance on predefined logical rules can be limiting, as constructing such rules could be challenging or domain-specific.
+ The advantage of using a factor graph over simpler methods (e.g., directly detecting inconsistencies) is not clearly justified.
+ The paper does not compare AGAIN with other methods that integrate prior knowledge for adversarial defense (e.g., DeepProblog).
+ Scalability issues arise from manually annotated concepts, large factor graphs, and computational overhead from exploring all intervention options.
+ Some sections, such as implementation details and explanations of figures (e.g., Figure 3 and Figure 7), lack clarity.
+ The datasets used may not reflect real-world adversarial scenarios, limiting the method's general applicability.

Most concerns have been addressed by the authors during the rebuttal period.

**Additional Comments On Reviewer Discussion:**

This paper ended up with ratings 8, 6, 6, 6, 5. The only negative reviewer’s major concerns are “relying on pre-defined concepts and logics.” The authors clarified that “how to automatically extract rules and concepts is not the focus of this paper but rather another problem worth investigating,” which I tend to agree to. There are a number of works that rely on pre-defined concepts. I therefore recommend accepting this paper.

---

### Decision · Program_Chairs · 2025-01-22

Accept (Poster)